# Inverse Reinforcement Learning Using Just Classification and a Few Regressions

## Abstract

Inverse reinforcement learning (IRL) aims to explain observed behavior by uncovering an underlying reward. In the maximum-entropy or Gumbel-shocks-to-reward frameworks, this amounts to fitting a reward function and a soft value function that together satisfy the soft Bellman consistency condition and maximize the likelihood of observed actions. While this perspective has had enormous impact in imitation learning for robotics and understanding dynamic choices in economics, practical learning algorithms often involve delicate inner-loop optimization, repeated dynamic programming, or adversarial training, all of which complicate the use of modern, highly expressive function approximators like neural nets and boosting. We revisit softmax IRL and show that the population maximum-likelihood solution is characterized by a *linear* fixed-point equation involving the behavior policy. This observation reduces IRL to two off-the-shelf supervised learning problems: probabilistic classification to estimate the behavior policy, and iterative regression to solve the fixed point. The resulting method is simple and modular across function approximation classes and algorithms. We provide a precise characterization of the optimal solution, a generic oracle-based algorithm, finite-sample error bounds, and empirical results showing competitive or superior performance to MaxEnt IRL.

## 1 Introduction

Behavioral data are abundant in robotics, human–computer interaction, healthcare, and economics. Inverse reinforcement learning (IRL) aims to determine the underlying and unobserved rewards that rationalize this behavior. Doing so provides a useful lens for explaining observed behavior, learning system structure, optimizing algorithmic policies, or choosing changes to the system that induce more preferable behavior. Overall, IRL is a key approach to gleaning generalizable insights from purely behavioral data, but it is not always so easy to operationalize.

Maximum entropy (MaxEnt) IRL is an especially appealing framework where the behaving agent is assumed to maximize cumulative rewards plus a bonus for policy entropy (Ziebart et al., 2008). It is equivalent to dynamic discrete choice (DDC) models assuming a rational agent facing random Gumbel-distributed shocks to immediate rewards (Rust, 1987). This induces softmax-like behavior, smoothing out otherwise brittle argmax behavior and allowing to explain observed behavior that may not always be exactly optimal. Yet, many DDC or MaxEnt IRL algorithms that realize this elegant theory are difficult to scale, generalize, and tune: nested optimization problems, restrictions to linear reward models, sensitivity to initialization and step sizes, and complex adversarial training loops.

Much of this complexity arises from too-directly tackling the IRL problem: find the Markov decision process (MDP) that matches observed transitions, satisfies a reward normalization for identifiability, and maximizes likelihood of observed actions under the policy that maximizes reward (equivalently, mean reward plus entropy in the MaxEnt persepctive) in this MDP. Many algorithms directly optimize this problem (Audiffren et al., 2015; Levine et al., 2011; Snoswell et al., 2020; Ziebart et al., 2008). For example, Ziebart et al. (2008) use gradient ascent in this optimization problem using a linear parametrization of the reward. Other algorithms nest or alternate searching over MDPs and solving the RL problem implied by each (Fu et al., 2018; Ho & Ermon, 2016; Rust, 1987; Wulfmeier et al., 2016). For example, Rust (1987) nests a policy iteration loop inside a parameter search. (A complete literature review is given in section 6.)

In this paper, we instead first solve a simpler but highly under-specified optimization problem given by dropping the reward normalization. We show one trivial solution (among many optimal solutions) is the logarithm of the behavior policy. Moreover, all other solutions are given via a transformation due to the invariance of observed behavior to the introduction of a state-level potential function. Therefore, to solve the original problem, it remains to find the state potential function that would transform the trivial solution in order to also satisfy the previously-relaxed reward normalization. Overall, this is given by a linear integral equation involving the log behavior policy and suggests a simple algorithm: classify to learn the behavior policy and iterate a regression a few times (namely, $\log(n)$ times for $n$ observations) to solve the equation.

This yields a meta-algorithm with a simple implementation that just requires calling off-the-shelf supervised learning routines a few times. The approach is a model-free, completely avoiding parameterizing the MDP or imposing special structure like linear rewards, and permits flexible function approximation via flexible supervised learning subroutines, e.g., a neural net regression. Theoretically, we can characterize the statistical behavior for general function approximation and obtain bounds from assuming high-level PAC bounds on the supervised learning routines, which we can in particular instantiate for nonparametric least squares under a Bellman completeness assumption on the hypothesis class and functional complexity measures. Empirically, the the algorithm is simple and effective.

## 1.1 CONTRIBUTIONS

Our main technical and methodological contributions are:

1. **Linear characterization via a normalization equation.** We show that all maximum-likelihood solutions correspond to potential-based reward shapings of $(r, v) = (\log \pi, 0)$, where $\pi(a \mid s)$ is the behavior policy. Imposing a per-state normalization yields a unique solution characterized by a simple *linear* equation in a state potential.

2. **A generic two-oracle algorithm.** Given access to any probabilistic classifier for $\pi(a \mid s)$ and any regression routine for conditional expectations, a short fitted fixed-point loop solves the normalization equation and outputs estimates $(\hat{r}, \hat{v})$.

3. **Finite-sample guarantees.** We establish a data-free oracle inequality that controls the error of approximate fixed-point iterates in terms of classification and regression inaccuracies. We then derive high-probability (finite-sample) guarantees via sample-splitting, showing that the estimation error of $(\hat{r}, \hat{v})$ decays at the statistical rates of the classifier and regressor.

## 2 PROBLEM SET UP

We review the softmax IRL setting from both structural discrete choice and maximum-entropy viewpoints, then formalize the optimization problem central to our analysis. We observe data $\{(s_i, a_i, s_i')\}_{i=1}^n$ of state–action-next-state transitions sampled from a distribution $P$ representing the observed behavior of an agent. The action space $\mathcal{A}$ is assumed finite ($|\mathcal{A}| < \infty$), and the state space $\mathcal{S}$ is a measurable spaces and can each be discrete or continuous (in our paper the complexity/cardinality of both spaces is captured wholly through the complexity of hypothesis classes of functions on them).

The goal of IRL is to infer a reward function $r : \mathcal{S} \times \mathcal{A} \to \mathbb{R}$ consistent with the apparent behavior policy $\pi(a \mid s) = P(a \mid s)$ being optimal in some sense in the corresponding MDP with a given discount factor $\gamma \in [0, 1)$. The data specifies the transition distribution $P(s' \mid s, a)$. What remains to specify an MDP is a reward distribution.

We now discuss in what sense $\pi$ should be optimal in the recovered MDP. One perspective is that of DDC (Aguirregabiria & Mira, 2010; Hotz & Miller, 1993; Rust, 1987). In these econometric models of sequential discrete decisions, at each time $t$, an agent sees rewards $r(s_t, a) + \varepsilon_t(a)$ for each action $a$, where $\varepsilon_t(a_t)$ are mean-zero idiosyncratic shocks (of known distribution) to the (unknown and to-be-inferred) mean reward function $r(s_t, a_t)$. In this setting, the value function is

$$V(s) = \mathbb{E}\left[\sum_{t \geq 0} \gamma^t (r(s_t, a_t) + \varepsilon_t(a_t)) \mid s_0 = s\right],$$

where expectation is over trajectories $s_0, a_0, s_1, a_1, \ldots$ with distribution $\pi(a_0 \mid s_0) P(s_1 \mid a_0, s_0) \pi(a_1 \mid s_1) \cdots$ and independent reward shocks $\varepsilon_t(a)$. We can also define

$$v(s, a) = PV(s, a), \quad Q(s, a) = r(s, a) + \gamma v(s, a),$$

using the shorthand that, for any state function $f(s)$ we define

$$Pf(s, a) = \mathbb{E}_{s' \sim P(\cdot \mid s, a)}[f(s')] = \int f(s') P(ds' \mid s, a).$$

Another handy shorthand we will use is that, for any state-action functions $f(s, a)$ and $\mu(a \mid s)$ (usually a policy), we define the $\mu$-expectation and the log-sum-exp over actions, repsectively, as

$$\mu f(s) = \sum_a \mu(a \mid s) f(s, a), \quad \Xi f(s) = \log(\sum_a \exp(f(s, a))).$$

The rational agent maximizes the value function from any starting point. When $\varepsilon_t(a_t)$ are chosen to be i.i.d. Gumbel (type-I generalized extreme value distribution), the resulting rational behavior is exactly to softmax the $Q$-function: $\pi^*(a \mid s) \propto e^{Q(s,a)}$. And, averaging over the Gumbel idyosyncracies that will get maxed, the reward to go from state $s$ is $\Xi Q(s)$. Thus, we obtain the **soft Bellman equation**: $v = P\Xi(r + \gamma v)$. Or, written out:

$$v(s, a) \;=\; \int \log \sum_{a'} \exp(r(s', a') + \gamma \, v(s', a')) \, P(ds' \mid s, a) \quad \forall s, a.$$

A wholly equivalent perspective is given by MaxEnt IRL (Ziebart et al., 2008; 2010). Here, one assumes that the agent seeks to maximizes the cumulative long-run discounted-average rewards plus a bonus for the entropy of the policy. This also leads to the same $\pi^*$ being optimal. And, the rewards plus entropy bonus from state $s$ onward is again $\Xi Q(s)$, leading to the same soft Bellman equation.

In view of this, the aim is then to find rewards that induce an MDP that best matches the data in terms of the (average conditional) **likelihood** assigned by the induced $\pi^*$ to observing the behavior we in fact see, of playing $a_i$ in state $s_i$ (given $s_i$ and averaging over it). In other words, minimize the Kullback-Leibler (KL) divergence between $\pi(\cdot \mid s)$ and $\pi^*(\cdot \mid s)$, averaged over states $s \sim P$.

While this is of course optimized when $\pi^* = \pi$, it is insufficient to determine rewards: the softmax policy is invariant to translating the $Q$-function by an arbitrary state-dependent function. To resolve this ambiguity, we must impose some reward normalization. For example, in discrete choice modeling (dynamic or otherwise), to anchor things, it is often assumed that a specific action has zero reward (e.g., the "outside option" of not choosing anything on offer in the context of a product offering) or alternatively that the sum (equivalently, uniform-weighted mean) of rewards is zero. Inferred rewards are understood as relative. In MaxEnt IRL it is common to anchor the value of the behavior policy. In this paper, we consider a general reward normalization requiring the reward to integrate to zero against a reference conditional measure $\mu(\cdot \mid s)$ (e.g., a point mass at a reference action, a uniform measure, or the behavior policy). That is, we enforce $\sum_a \mu(a \mid s) r(s, a) = 0$ for all $s$, or $\mu r = 0$

**Main problem:** Put together, the main problem of interest is to maximize over state-action functions $r, v$ the conditional log likelihood, subject to soft Bellman and reward normalization:

$$\arg\max_{r,v} \quad \mathbb{E}_{(a,s) \sim P}[r(s, a) + \gamma \, v(s, a) - \log \sum_{a'} \exp(r(s, a') + \gamma \, v(s, a'))]$$
$$\text{s.t.} \quad v = P\Xi(r + \gamma v) \qquad \text{(soft Bellman)}, \tag{1}$$
$$\mu r = 0 \qquad\qquad\qquad \text{(reward normalized)}$$

The remainder of the paper shows that solving (1) reduces to first finding $\pi$ and then solving a linear equation to find a state potential function, and then operationalizing this idea in a simple algorithm.

## 3 CHARACTERIZATION VIA A RELAXED PROBLEM

To highlight the underlying structure, we first drop the normalization and analyze the relaxed problem.

**Relaxed problem:** We relax reward normalization in eq. (1) but still enforce soft Bellman consistency:

$$\arg\max_{r,v} \quad \mathbb{E}_{(a,s) \sim P}[r(s, a) + \gamma \, v(s, a) - \log \sum_{a'} \exp(r(s, a') + \gamma \, v(s, a'))]$$
$$\text{s.t.} \quad v = P\Xi(r + \gamma v) \qquad \text{(soft Bellman)} \tag{2}$$

This optimization problem is highly under-specified (has many solutions) because the likelihood objective is flat along any state-potential reshaping of the reward. This is the exact invariance we will use to back out a solution to eq. (1), after finding just one easy-to-identify optimal solution.

### 3.1 A TRIVIAL SOLUTION TO THE RELAXED PROBLEM

When the normalization is absent, there is a trivial but informative solution: set the reward to the log behavior policy, $r(s, a) = \log \pi(a \mid s)$, and the soft action-value function to zero, $v(s, a) = 0$. This choice exactly maximizes the conditional log-likelihood and satisfies the soft Bellman relation, since the next-state log-partition vanishes under the policy normalization ($\sum_a \exp(r(s, a)) = 1$).

In the rest of the paper we will define the log-behavior-policy as

$$u^\star(s, a) = \log \pi(a \mid s).$$

**Lemma 1** (Trivial optimum for the relaxed problem). $(r, v) = (u^\star, 0)$ *is an optimal solution to* (2).

In the next subsection, we show that this trivial solution already suffices to compare policy values.

### 3.2 ASIDE: POLICY COMPARISON

Any feasible solution to eq. (2), including the trivial one $(u^\star, 0)$, suffices for one of IRL's most fundamental tasks: comparing policies. The following theorem establishes that policy value differences are invariant to potential-based shaping, and hence constant over the entire solution set of eq. (2).

Given a reward $r$, the $Q$-function of a policy $\pi_1$ is defined by the fixed point $Q_r^{\pi_1} = r + \gamma \pi_1 P Q_r^{\pi_1}$. The policy's value function is $V_r^{\pi_1} = \pi_1 Q_r^{\pi_1}$, and we are often interested in comparing values of policies.

**Theorem 1** (Identification of policy value differences). *Let* $(r, v)$ *be an optimal solution to eq.* (2). *Then, for any policy* $\pi_1$,

$$Q_{u^\star}^{\pi_1}(s, a) \;=\; Q_r^{\pi_1}(s, a) - \Xi(r + \gamma v)(s).$$

*Consequently, for any two policies* $\pi_1, \pi_2$,

$$V_{u^\star}^{\pi_1}(s) - V_{u^\star}^{\pi_2}(s) \;=\; V_r^{\pi_1}(s) - V_r^{\pi_2}(s).$$

This identification mirrors the central observation of Hotz & Miller (1993): differences in action-specific value functions are revealed by log-odds of the observed behavior policy (for Gumbel idiosyncratic shocks to rewards). In our setting, $u(s, a) = \log \pi(a \mid s)$ plays the same role, showing that the trivial solution suffices for recovering policy value differences.

Nonetheless, precise reward recovery is required when evaluating policies under different dynamics (e.g., shifting $P$), different time horizons (e.g., changing $\gamma$ for long-term vs. short-term planning), or when estimating structural parameters of $r$ and $v$ in economic models. In such cases, solving the normalization-constrained problem eq. (1) is essential to identify the true reward, which is the focus of the remainder of the paper (and, differently from Hotz & Miller, 1993; Hotz et al., 1994, we will fit $v$ directly and possibly non-parametrically using function approximators, rather than use simulation to infer moments on a parametrization of $r$ that would be solved by a method of moments).

### 3.3 AN INVARIANCE AMONG SOLUTIONS

The relaxed problem is invariant to *potential-based shaping*: adding a state-only potential $c : \mathcal{S} \to \mathbb{R}$ shifts all logits in the softmax by the same amount per state, leaving both feasibility and likelihood unchanged. This is the entropy-regularized analogue of reward shaping in classical RL (Ng et al., 1999) and explains why the relaxed objective is flat along an affine subspace.

**Lemma 2** (Potential-based shaping invariance). *Let* $(r, v)$ *be feasible in eq.* (2) *(i.e., satisfies soft Bellman) and let* $c : \mathcal{S} \to \mathbb{R}$ *be arbitrary. Define* $\tilde{r} = r + c - \gamma Pc$, $\tilde{v} = v + Pc$, *or explicitly*

$$\tilde{r}(s, a) = r(s, a) + c(s) - \gamma \int c(s') P(ds' \mid s, a), \quad \tilde{v}(s, a) = v(s, a) + \int c(s') P(ds' \mid s, a).$$

*Then* $(\tilde{r}, \tilde{v})$ *is also feasible in eq.* (2), *where it obtains the same objective value as* $(r, v)$.

### 3.4 SOLVING THE ORIGINAL (NORMALIZATION-CONSTRAINED) PROBLEM

Lemma 1 gives us one optimal solution to the relaxed problem, eq. (2), and lemma 2 gives us a way to transform it to obtain a variety of other also-optimal solutions. If eq. (1) and eq. (2) have the same

optimal value (which they do), then optimizers of eq. (1) are the optimizers of eq. (2) that satisfy reward normalization. What remains in order to solve the original problem, (1), is to transform the trivial solution until we also satisfy the reward normalization, which amounts to a linear equation in the state potential.

**Theorem 2** (IRL as a linear equation). *Equation* (1) *admits a unique optimal solution* $r^\star, v^\star$, *where* $r^\star = u^\star - v^\star + \mu(\gamma v^\star - u^\star)$ *and* $v^\star$ *is the unique bounded solution to the fixed point*

$$v^\star = P\mu(\gamma v^\star - u^\star),$$

*or, written explicitly,* $v^\star(s,a) = \mathbb{E}_{s' \sim P(\cdot|s,a)}[\sum_{a'} \mu(a' \mid s')(\gamma v^\star(s',a') - u^\star(s',a')) \mid s,a] \ \ \forall s,a$, *and* $r^\star(s,a) = u^\star(s,a) - v^\star(s,a) + \sum_{a'} \mu(a' \mid s)(\gamma v^\star(s,a') - u^\star(s,a'))$.

The solutions $r^\star, v^\star$ can also be written in the form of lemma 2 as $r^\star = u^\star + c^\star - \gamma P c^\star, v^\star = P c^\star$, where the "right" state potential $c^\star : \mathcal{S} \to \mathbb{R}$ is the unique bounded solution to $c^\star - \gamma\mu P c^\star = -\mu u^\star$.

This theorem is the fulcrum of the paper. It reduces IRL to computing $u^\star = \log \pi$ and then solving a linear equation (involving $u^\star$) for $c^\star$. The reward and soft value functions are then given by these alone. We next show how to operationalize this with a minimal algorithm.

## 4 A GENERIC ALGORITHM

The solution characterization suggests a two-oracle procedure: learn $\hat{u}$ by probabilistic classification, then solve for $\hat{v}$ by fitted fixed-point regression. This is described in algorithm 1 below.

---

**Algorithm 1** CLASSIFY-THEN-REGRESS IRL: a simple IRL solver with general function approximation via blackbox classification and regression oracles

---

1: **Inputs:** transitions $\{(s_i, a_i, s_i')\}_{i=1}^n$; reference measure $\mu(a \mid s)$; discount $\gamma$; a classification algorithm; a regression algorithm
2: **Classify:** fit $\hat{\pi}(a \mid s)$ by classifying $y_i = a_i$ given $x_i = s_i$
3: $\hat{u}(s,a) \leftarrow \log \hat{\pi}(a \mid s)$
4: Initialize $\hat{v}^{(0)}(s,a) = 0$
5: **for** $k = 1, \ldots, K$ **do**
6:     **Regress:** fit $\hat{v}^{(k)}$ by regressing $y_i = \sum_a \mu(a \mid s_i')\big(\gamma\,\hat{v}^{(k-1)}(s_i',a) - \hat{u}(s_i',a)\big)$ on $x_i = (s_i, a_i)$
7: **end for**
8: **Return** $\hat{r}(s,a) = \hat{u}(s,a) + \sum_{a'} \mu(a' \mid s)\big(\gamma\,\hat{v}^{(K)}(s,a') - \hat{u}(s,a')\big) - \gamma\hat{v}^{(K)}(s,a), \ \ \hat{v} = \hat{v}^{(K)}$

---

We can understand the algorithm as approximating a fixed point iteration. Define for any $u$ the map

$$T_u v = P\mu(\gamma v - u).$$

Then, $v^\star$ is the unique fixed point of $T_{u^\star}$. We approximate $T_{u^\star}$ by plugging in $\hat{u}$ and by replacing the map $P$ with a regression algorithm. We then iterate this approximation a few times to obtain $\hat{v}$. As we will argue in the next section, only a modest number of iterations (roughly $K \approx \log n$) are needed for the iteration error to be negligible relative to statistical error. This is because the operator $T_u$ is a $\gamma$-contraction and therefore fixed point iterations converge exponentially fast.

Our procedure is deliberately blackbox: any calibrated classifier and regressor can be used in practice. For example, one may use neural networks or gradient boosting with cross-entropy loss for classification and with mean-squared error for regression. Practically, the regression step also need not be solved fully. We could, as in Deep $Q$-Networks (Mnih et al., 2013), parametrize $\hat{v}$ by a neural net and, for each "**Regress**" step, only take a single (or more) stochastic gradient step in the mean-squared error loss before updating the target network $\hat{v}$ for the next gradient step.

## 5 THEORETICAL GUARANTEES

This section develops a two-layer analysis of the proposed procedure. We first present a *deterministic* (data-free) oracle inequality that bounds the error of any $K$-step approximate fixed-point iterate in terms of two primitive quantities: the input mismatch $\nu := \|\hat{u} - u^\star\|_2$ (the quality of the classification step) and the per-iteration inexactness $\eta_k := \|\hat{v}^{(k)} - T_{\hat{u}}\hat{v}^{(k-1)}\|_2$ (the quality of each regression

step). Having established this, we then provide *high-probability* instantiations of these quantities via sample-splitting: $\hat{u}$ is learned on one half of the data, and the $K$ regression steps are performed on $K$ disjoint subfolds of the other half to obtain $\{\hat{v}^{(k)}\}_{k=1}^{K}$.

**Notations for this section.** In the following, let $v_u$ be the unique bounded fixed point of $T_u$, which exists because $T_u$ is an affine $\gamma$-contraction in the supremum norm. Note that $v^\star = v_{u^\star}$. Let $\lambda$ be a state distribution satisfying the stationary $\lambda P\mu = \lambda$. We write $\|c\|_{2,\lambda\otimes\mu} := (\mathbb{E}_{s\sim\lambda}[c(s)^2])^{1/2}$, $\|g\|_{2,\mu\otimes\lambda} := (\mathbb{E}_{(s,a)\sim\lambda\otimes\mu}[g(s,a)^2])^{1/2}$, and $\|g\|_2 := (\mathbb{E}_{(s,a)\sim P}[g(s,a)^2])^{1/2}$. We assume that the density ratio $\frac{\lambda\otimes\mu}{P}(a,s)$ is uniformly bounded with $\kappa := \operatorname{ess\,sup}_{s,a}\frac{(\lambda\otimes\mu)(s,a)}{P(s,a)} < \infty$, so that $\|\cdot\|_{2,\lambda\otimes\mu} \lesssim \|\cdot\|_2$. All subsequent results remain valid if $\|\cdot\|_{2,\lambda\otimes\mu}$ and $\|\cdot\|_{2,\lambda\otimes\mu}$ are replaced by the supremum norms.

## 5.1 GENERIC ANALYSIS OF APPROXIMATE FIXED-POINT ITERATES

We begin by isolating two structural facts about the operator $T_u$ and its fixed points. The first shows that the map $u \mapsto v_u$ is Lipschitz. The second quantifies how inexact iterations accumulate error.

**Lemma 3** (Lipschitz stability of the fixed-point map). *Let $v_u$ and $v_{u'}$ be the unique fixed points of $T_u$ and $T_{u'}$, respectively. Then*

$$\|v_u - v_{u'}\|_{2,\lambda\otimes\mu} \leq \tfrac{1}{1-\gamma}\|u - u'\|_{2,\lambda\otimes\mu}.$$

We next quantify the impact of per-iteration inexactness when iterating $A_{\hat{u}}$.

**Lemma 4** (Inexact iterations of $T_{\hat{u}}$). *Fix $\hat{u}$. Consider any sequence $\{\hat{v}^{(k)}\}_{k=0}^{K}$ such that, for $k \geq 1$,*

$$\|\hat{v}^{(k)} - T_{\hat{u}}\hat{v}^{(k-1)}\|_{2,\lambda\otimes\mu} \leq \eta_k.$$

*Then, $\|v_{\hat{u}} - \hat{v}^{(K)}\|_{2,\lambda\otimes\mu} \leq \gamma^K\|v_{\hat{u}} - \hat{v}^{(0)}\|_{2,\lambda\otimes\mu} + \sum_{t=1}^{K}\gamma^{K-t}\eta_t.$*

Combining the two lemmas with the triangle inequality yields a deterministic bound in terms of input error (classification error) and iteration error (regression error).

**Theorem 3** (Deterministic inequality for the $K$-step iterate). *Fix $\hat{u}$ satisfying error $\nu := \|\hat{u} - u^\star\|_{2,\lambda\otimes\mu}$ and a sequence $\{\hat{v}^{(k)}\}_{k=0}^{K}$ with errors $\eta_k := \|\hat{v}^{(k)} - T_{\hat{u}}\hat{v}^{(k-1)}\|_{2,\lambda\otimes\mu}$. Then*

$$\|\hat{v}^{(K)} - v^\star\|_{2,\lambda\otimes\mu} \leq \tfrac{1}{1-\gamma}\nu + \gamma^K\|v_{\hat{u}} - \hat{v}^{(0)}\|_{2,\lambda\otimes\mu} + \sum_{k=1}^{K}\gamma^{K-k}\eta_k.$$

The next lemma shows that this $L^2(\lambda\otimes\mu)$ control suffices to control the reward error $\|\hat{r} - r^\star\|_2$.

**Lemma 5.** *Let $\operatorname{ess\,sup}_{s,a}\frac{P(s,a)}{\lambda(s)\mu(a|s)} < \infty$. Then, $\|\hat{r}^{(K)} - r^\star\|_2 \lesssim \|\hat{u} - u\|_2 + \|\hat{v}^{(K-1)} - v^\star\|_{2,\lambda\otimes\mu} + \eta_K$.*

## 5.2 HIGH-PROBABILITY INSTANTIATION VIA SAMPLE SPLITTING

We now instantiate the deterministic oracle inequality using a sample-splitting scheme and generic supervised learning PAC guarantees. The $n$ samples are divided into two halves. On the first half, we estimate $\hat{u} = \log\hat{\pi}$ by probabilistic classification over a function class $\mathcal{U} \subseteq \{u : \mathcal{S}\times\mathcal{A}\to\mathbb{R}\}$ of log-policies. On the second half, we further partition the data into $K$ disjoint folds of size $\lfloor n/(2K)\rfloor$. On each fold, we fit the value estimate $\hat{v}^{(k)}$ by least-squares regression of $\mu(\gamma\hat{v}^{(k-1)} - \hat{u})(s_i')$ on $(s_i, a_i)$ over a convex class $\mathcal{V} \subseteq \{v : \mathcal{S}\times\mathcal{A}\to\mathbb{R}\}$, and then compute $\hat{c}^{(k)}$ via a Bellman-like update. This structured specialization of Algorithm 1 is summarized in Algorithm 2 in Appendix A, and Lemma 6 provides the corresponding high-probability bounds for $\|\hat{u} - u^\star\|_2$ and the per-iteration inexactness $\|\hat{v}^{(k)} - T_{\hat{u}}\hat{v}^{(k-1)}\|_2$.

To study generalization, assume both $\mathcal{U}$ and $\mathcal{V}$ are uniformly bounded by some constant $B < \infty$, which ensures that $\hat{u}$ and $\hat{v}$ are bounded. We assume that the fitted policy $\hat{\pi}$ input to Alg. 2 satisfies a PAC-type guarantee with respect to a (possibly data-dependent) function $\hat{\rho}_{\mathcal{U}}\left(\frac{n}{2}, \delta\right)$ (see, e.g., van de Geer, 2000 Chapter 7 for such guarantees on cross entropy over a generic nonparmateric function class of conditional probabilities).

**Assumption 1** (Policy generalization). *For all $\delta \in (0, 0.5)$, $\hat{u}, u \in \mathcal{U}$ and*

$$\mathbb{E}_{s \sim P}\big[\mathrm{KL}\big(\pi(\cdot \mid s) \,\|\, \hat{\pi}(\cdot \mid s)\big)\big] \ \leq \ \{\hat{\rho}_{\mathcal{U}}\big(\tfrac{n}{2}, \delta\big)\}^2 \quad \textit{with probability at least } 1 - \delta.$$

The sample-split least-squares instantiation enables a direct application of standard learning theory to obtain a generalization bound for $\hat{v}^{(k)}$ as an estimator of $T_{\hat{u}}\hat{v}^{(k-1)}$. For a sample of size $n$ and radius $r > 0$, the localized empirical Rademacher complexity of $\mathcal{V}$ is defined as $\hat{\mathfrak{R}}_n(\mathcal{V}, r) :=$ $\mathbb{E}_{\varepsilon}\Big[\sup_{v,w \in \mathcal{V}: \|v - w\|_2 \leq r} \frac{1}{n} \sum_{i=1}^{n} \varepsilon_i (v - w)(s_i, a_i)\Big]$ where $\varepsilon_i \overset{\text{i.i.d.}}{\sim} \mathrm{Rad}(\pm 1)$. The PAC-type bound we will prove for $\hat{v}^{(k)}$ is expressed in terms of the empirical critical radius:

$$\hat{\rho}_{\mathcal{V}}(n, \delta) := \hat{r}_{\mathcal{V}}(n) + \sqrt{\tfrac{\log(1/\delta)}{n}}, \quad \hat{r}_{\mathcal{V}}(n) := \inf\{r > 0 : \hat{\mathfrak{R}}_n(\mathcal{V}, r) \lesssim r^2\},$$

with constants depending only on $B$. Finally, we assume correct specification of the function classes through Bellman completeness.

**Assumption 2** (Bellman completeness). *For any $u \in \mathcal{U}$ and $v \in \mathcal{V}$, one has $T_u v \in \mathcal{V}$.*

Under Assumption 2, each regression target $T_{\hat{u}}\hat{v}^{(k-1)}$ belongs to $\mathcal{V}$. Thus, nonparametric least squares incurs estimation but no approximation error on each fold. The following lemma collects the resulting high-probability bounds.

**Lemma 6** (High-probability events for least-squares classification and $K$ regressions). *Suppose Algorithm 2 is executed (i.e., split the sample for classification and each regression step). Then, for any $\delta \in (0, 1)$, there exists a constant $C = C(B)$ such that, with probability at least $1 - \delta$,*

$$\|\hat{u} - u\|_2 \ \leq \ C\,\hat{\rho}_{\mathcal{U}}\big(\tfrac{n}{2}, \tfrac{\delta}{2}\big), \qquad \|\hat{v}^{(k)} - T_{\hat{u}}\hat{v}^{(k-1)}\|_2 \ \leq \ C\,\hat{\rho}_{\mathcal{V}}\Big(\big\lfloor \tfrac{n}{2K} \big\rfloor, \tfrac{\delta}{2K}\Big).$$

Plugging these events into the deterministic oracle inequality gives the desired finite-sample guarantee.

**Theorem 4** (Sample complexity for the $K$-step iterate). *Let $\kappa := \operatorname{ess\,sup}_{(s,a)} \frac{d(\lambda \otimes \mu)}{dP}(s, a) < \infty$. Under Assumption 2, for any $\delta \in (0, 1)$ there exists $C = C(B, \kappa)$ such that, with probability at least $1 - \delta$, the $K$-step output satisfies*

$$\|\hat{v}^{(K)} - v^{\star}\|_{2, \lambda \otimes \mu} \ \leq \ \frac{1}{1 - \gamma} C\,\hat{\rho}_{\mathcal{U}}\big(\tfrac{n}{2}, \tfrac{\delta}{2}\big) + \gamma^K \|v_{\hat{u}} - \hat{v}^{(0)}\|_{2, \lambda \otimes \mu} + \frac{1 - \gamma^K}{1 - \gamma} C\,\hat{\rho}_{\mathcal{V}}\Big(\big\lfloor \tfrac{n}{2K} \big\rfloor, \tfrac{\delta}{2K}\Big).$$

*In particular, taking $K \simeq c \log n$ (so that $\gamma^K$ is negligible) yields*

$$\|\hat{v}^{(K)} - v^{\star}\|_{2, \lambda \otimes \mu} \ \lesssim \ \frac{1}{1 - \gamma}\Big\{\hat{\rho}_{\mathcal{U}}\big(\tfrac{n}{2}, \tfrac{\delta}{2}\big) + \hat{\rho}_{\mathcal{V}}\Big(\big\lfloor \tfrac{n}{2K} \big\rfloor, \tfrac{\delta}{2K}\Big)\Big\}.$$

The theorem reduces statistical analysis of the algorithm to well-understood localized complexities for the chosen function classes. For instance, if $\mathcal{U}$ and $\mathcal{V}$ are parametric with pseudo-dimensions $d_{\mathcal{U}}$ and $d_{\mathcal{V}}$, then typically $\hat{\rho}_{\mathcal{U}}(m, \delta), \hat{\rho}_{\mathcal{V}}(m, \delta) \asymp \sqrt{(d + \log(1/\delta))/m}$, and the bound becomes

$$\|\hat{v}^{(K)} - v^{\star}\|_{2, \lambda \otimes \mu} \ \lesssim \ \frac{1}{1 - \gamma}\left(\sqrt{\frac{d_{\mathcal{U}} + \log(1/\delta)}{n}} + \sqrt{\frac{\log n\,(d_{\mathcal{V}} + \log(\log n/\delta))}{n}}\right).$$

# 6 RELATED LITERATURE

We expand on connections to structural econometrics, IRL/imitation, and entropy-regularized control.

**Structural dynamic discrete choice (DDC).** The nested fixed point approach of Rust (1987) and the two-step estimator of Hotz & Miller (1993) underlie much of empirical dynamic choice. Identification in DDC with unobserved heterogeneity and exclusion restrictions is studied by Magnac & Thesmar (2002). Surveys by Aguirregabiria & Mira (2010) and Arcidiacono & Ellickson (2011) detail computational and statistical strategies. Our work is closest to softmax DDC with Gumbel shocks, where the choice probabilities have a logit form and the soft Bellman equation governs continuation values. The novelty here is to side-step nested policy iterations, simulation procedures, and generalized methods of moments, and instead leverage generic supervised learning for nonparametric, flexible estimation using a fitted iteration akin to model-free value-based reinforcement learning.

**Classical IRL and MaxEnt IRL.** Foundational IRL work framed reward recovery as feature expectation matching (Abbeel & Ng, 2004; Ng & Russell, 2000). Maximum (causal) entropy IRL (Ziebart, 2010; Ziebart et al., 2008) introduced a probabilistic foundation with convexity and a soft Bellman consistency. Relative-entropy and maximum-margin variants (Ratliff et al., 2006) provided alternative regularizations and learning signals. Bayesian IRL (Ramachandran & Amir, 2007) quantified uncertainty at the cost of heavier computation. Deep IRL extensions include GP-based nonlinear rewards (Levine et al., 2011) and deep energy-based models (Wulfmeier et al., 2016). Many related works focus on imitation: GAIL (Ho & Ermon, 2016) casts imitation as occupancy measure matching via GAN training; AIRL (Fu et al., 2018) recovers a reward up to shaping by jointly training a discriminator and policy. These methods scale well but bring adversarial optimization challenges and do not directly target IRL for recovering underlying reward structure. Our work shows how to characterize the structural maximum-likelihood solution as a solution to an affine fixed point and leverage supervised learning with off-the-shelf generalization guarantees.

**Entropy-regularized RL and control as inference.** Linearly solvable MDPs and path-integral control (Kappen, 2005; Todorov, 2009) connect entropy-regularization to tractable value computations. Path-consistency learning (Nachum et al., 2017) enforces soft Bellman identities along trajectories, and SAC (Haarnoja et al., 2018) brings maximum entropy to off-policy actor–critic learning. Uehara et al. (2023) study entropy-regularized offline RL and show how slowly tempering regularization yields strong guarantees for vanilla offline RL. Levine (2018) surveys RL-as-inference, situating many of these methods under a common umbrella. Our work, like MaxEnt IRL, is complementary to this: it targets the *inverse* problem.

**Value-based offline RL.** Fitted $Q$-iteration (FQI) is a standard algorithm for offline RL that uses a regression oracle to approximate and iterate the Bellman optimality operator (Ernst et al., 2005; Mnih et al., 2013; Munos, 2005). Aside from our handling of the inexactness in $\hat{u}$, our analysis of the inexact fixed point iteration steps is overall similar to the approaches in analyses of FQI (Fan et al., 2020; Hu et al., 2025; Munos & Szepesvári, 2008). This literature also reveals the minimality of Bellman completeness to avoid suffering exponential-in-horizon terms (Amortila et al., 2020; Foster et al., 2021; Wang et al., 2021), which strongly suggests it is similarly unavoidable in our value-based IRL setup without another assumption in its place, as well as possible remedies such as representation learning (Chang et al., 2022; Pavse et al., 2024) or adversarial estimators that replace Bellman completeness with an assumption of rich-enough critic class (Chen & Jiang, 2019; Uehara et al., 2020; 2021).

## 7 EXPERIMENTS

We aim to show that accurate reward recovery (up to normalization) and expert-level policies can be obtained without complicated optimization, matching MaxEnt IRL when the function class is correctly specified and improving upon it when generic function approximation is required.

We evaluate our approach in three gridworld domains with discounted demonstrations ($\gamma = 0.97$). We compare (i) MaxEnt IRL with gradient-based reward fitting, and (ii) Ours, which first estimates the trivial solution $u = \log \pi$ using a simple neural classifier, and then computes the normalized solution by solving the corresponding fixed-point equation. For the MaxEnt baseline we implement differentiable soft value iteration (temperature $= 1$), trained with Adam, learning-rate scheduling, gradient clipping, and early stopping. Existing implementations in the `imitation` package and in `qzed/irl-maxent` only support action-independent rewards and were not applicable. All methods use the true transition kernel. Reward recovery is evaluated using Q-differences $Q(s, a) - Q(s, 0)$, which are invariant to normalization and isolate recovery up to potential shaping (reported as RMSE and correlation). Policy quality is measured by KL divergence, total variation (TV), and top-1 accuracy. All results are averaged over 100 reruns.

**Easy identifiable.** On a $4 \times 4$ torus gridworld with linear rewards, both methods achieve near-perfect recovery (Corr $\geq 0.99$) and indistinguishable policies (KL $\leq 0.006$).

**Identifiable.** In an $8 \times 8$ gridworld with tabular linear rewards, MaxEnt IRL is competitive, but Ours yields lower error (RMSE 0.0086 vs. 0.1081) with equally accurate policies.

**Misspecified.** With nonlinear rewards on the same $8 \times 8$ gridworld, the linear MaxEnt baseline underfits (Corr 0.83, KL 0.049), while Ours with a neural policy head improves both recovery and policy matching (Corr 0.73, KL 0.0032, TV 0.0083).

Table 1: Results in three gridworld domains. Q-diff (RMSE), KL, and TV: lower is better. Corr and Top-1: higher is better. Entries are mean $\pm$ SE across reruns.

| Exp. | Method | RMSE ↓ | Corr ↑ | KL ↓ | TV ↓ | Top-1 ↑ |
|------|--------|--------|--------|------|------|---------|
| Easy | MaxEnt | $0.17 \pm 0.03$ | $\mathbf{0.996 \pm 0.005}$ | $0.0057 \pm 0.0007$ | $0.045 \pm 0.005$ | $\mathbf{1.00}$ |
| | Ours | $\mathbf{0.13 \pm 0.02}$ | $0.992 \pm 0.006$ | $\mathbf{0.0011 \pm 0.0002}$ | $\mathbf{0.016 \pm 0.002}$ | $\mathbf{1.00}$ |
| Ident. | MaxEnt | $0.27 \pm 0.04$ | $0.981 \pm 0.005$ | $0.0016 \pm 0.0005$ | $0.018 \pm 0.004$ | $0.86 \pm 0.07$ |
| | Ours | $\mathbf{0.017 \pm 0.003}$ | $\mathbf{0.999 \pm 0.000}$ | $\mathbf{0.0000}$ | $\mathbf{0.0016 \pm 0.0002}$ | $\mathbf{0.93 \pm 0.07}$ |
| Hard | MaxEnt | $0.83 \pm 0.02$ | $0.561 \pm 0.031$ | $0.029 \pm 0.002$ | $0.094 \pm 0.004$ | $0.50 \pm 0.03$ |
| | Ours | $\mathbf{0.23 \pm 0.01}$ | $\mathbf{0.980 \pm 0.001}$ | $\mathbf{0.0011 \pm 0.0001}$ | $\mathbf{0.018 \pm 0.001}$ | $\mathbf{0.90 \pm 0.02}$ |

## 8 Conclusion, extensions, and future work

We have shown how the softmax IRL problem can be reframed as finding the state potential function that transform a trivial $v$-free solution to satisfy the given reward normalization. This suggested a simple algorithm: fit the behavior policy to find this trivial solution, iterate a few regressions to make it match the reward normalization and hence solve the problem. The algorithm is simple to implement, leverage off-the-shelf supervised learning routines, and admits guarantees that allow flexible nonparametric function approximation. Beyond suggesting a particular algorithm, we hope this work provides a lens to better understand IRL: if observing an expert then logits of the behavior policy already must capture long-term value, the rest is just attributing it over time to fit constraints.

We here focused on Gumbel reward shocks (equivalently, entropy regularized agent) as it is the most common. An extension would be to generalize this to reward shocks with any given cumulative distribution function $F$. This change would propagate to appropriately updating the soft Bellman equation and the trivial $u^\star$ solution, but many of the primary insights and algorithmic structure would remain. Another possible extension is to replace the fixed point regression iteration with a minimax estimator, analogous to minimax $Q$-estimators in offline RL (Uehara et al., 2021). This can allow relaxing Bellman completeness assumptions by decoupling $\mathcal{V}$ from the critic class, so richer classes only improve performance. Finally, one may consider alternative reward normalizations. This would lead to a different constraint to pin down the right element in the equivalence class from lemma 2 of the optimal solution from lemma 1.

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

## A   INSTANTIATED ALGORITHM

Our high-probability analysis is based on the following algorithm.

## B   PROOFS

This appendix contains detailed proofs of all claims. Each proof is accompanied by comments on intuition and scope.

---

**Algorithm 2** Split–Classify–Regress IRL

---

1: **Inputs:** transitions $\{(s_i, a_i, s'_i)\}_{i=1}^n$; reference measure $\mu(\cdot \mid s)$; discount factor $\gamma$; value class $\mathcal{V}$; steps $K$; a probabilistic multiclass classifier (outputs $\hat{\pi}(a \mid s)$).

2: **Split:** partition the data into $\mathcal{D}^{\text{cls}}$ and $\mathcal{D}^{\text{reg}}$ (sizes $n/2$ each).

3: **Classification on $\mathcal{D}^{\text{cls}}$:** train the classifier on $\{(s_i, a_i) \in \mathcal{D}^{\text{cls}}\}$ to obtain $\hat{\pi}(a \mid s)$; set $\hat{u}(s, a) = \log \hat{\pi}(a \mid s)$.

4: **Folds for regression:** split $\mathcal{D}^{\text{reg}}$ into $K$ folds $\{\mathcal{D}_k\}_{k=1}^K$ of size $\left\lfloor \frac{n}{2K} \right\rfloor$.

5: Initialize $\hat{v}^{(0)}(s, a) = 0$.

6: **for** $k = 1, \ldots, K$ **do**

7:     **Regression on fold $k$:** $\hat{v}^{(k)} \in \arg\min\limits_{v \in \mathcal{V}} \sum\limits_{(s_i, a_i, s'_i) \in \mathcal{D}_k} \Big\{ \underbrace{\mu\big[\gamma \hat{v}^{(k-1)} - \hat{u}\big](s'_i)}_{\text{outcome } y_i} - v(s_i, a_i) \Big\}^2.$

8:     **Bellman update:** $\hat{c}_k(s) \leftarrow \mu\big[\gamma \hat{v}^{(k)} - \hat{u}\big](s).$

9: **end for**

10: **Return:** $\hat{r}(s, a) = \hat{u}(s, a) + \hat{c}_K(s) - \gamma \hat{v}^{(K)}(s, a), \quad \hat{v}(s, a) = \hat{v}^{(K)}(s, a).$

---

## B.1 PROOF OF LEMMA 1

We start with feasibility: with $u = \log P_{\text{data}}$, for any $s'$ we have $\sum_{a'} \exp\{u(s', a')\} = 1$, so the RHS of the soft Bellman equals 0 and $v \equiv 0$ is feasible. For optimality, write the per-sample log-likelihood

$$\ell(r, v; s, a) = r(s, a) + \gamma v(s, a) - \log \sum_{a'} e^{r(s,a') + \gamma v(s,a')}.$$

The right-hand side is the log of a categorical distribution $\pi(\cdot \mid s) \propto \exp\{r + \gamma v\}$, hence

$$\mathbb{E}\big[\ell(r, v; s, a)\big] = -\mathbb{E}_s \text{KL}\big(P_{\text{data}}(\cdot \mid s) \,\|\, \pi(\cdot \mid s)\big) + \mathbb{E}_s H\big(P_{\text{data}}(\cdot \mid s)\big),$$

maximized when $\pi(\cdot \mid s) = P_{\text{data}}(\cdot \mid s)$ for all $s$, which $(u, 0)$ achieves.

## B.2 PROOF OF LEMMA 2

Let $\tilde{r} = r + c - \gamma \int c(s') P(ds' \mid s, a)$ and $\tilde{v} = v + \int c(s') P(ds' \mid s, a)$. For any $s'$,

$$\sum_{a'} e^{\tilde{r}(s',a') + \gamma \tilde{v}(s',a')} = e^{c(s')} \sum_{a'} e^{r(s',a') + \gamma v(s',a')}.$$

Taking log and $\mathbb{E}[\cdot \mid s, a]$ shows the soft Bellman is preserved. For the objective, observe

$$\tilde{r}(s, a) + \gamma \tilde{v}(s, a) = r(s, a) + \gamma v(s, a) + c(s), \qquad \log \sum_{a'} e^{\tilde{r} + \gamma \tilde{v}} = c(s) + \log \sum_{a'} e^{r + \gamma v},$$

so the per-sample log-likelihood is unchanged.

## B.3 PROOF OF THEOREM 2

By Lemmas 1 and 2, all optima are $(u, 0)$ shaped by a potential $c$. Enforcing the gauge $\int r \, d\mu = 0$ yields

$$0 = \mu u(s) + c(s) - \gamma \int \int c(s') P(ds' \mid s, a) \, \mu(da \mid s) = \mu u(s) + c(s) - \gamma (P_\mu c)(s),$$

i.e., $(I - \gamma P_\mu) c = -\mu u$. Since $P_\mu$ is a Markov operator with sup-norm $\leq 1$ and $\gamma < 1$, $I - \gamma P_\mu$ is invertible on bounded functions via the Neumann series, giving a unique $c$ and hence unique $(r^\star, v^\star)$.

## B.4 PROOFS FOR SECTION 5

*Proof of Lemma 3.* Recall $\lambda$ is stationary for the state kernel $P_\mu(s, ds') := \int P(ds' \mid s, a) \, \mu(da \mid s)$, and let $\lambda \otimes \mu$ denote the induced stationary distribution on $(s, a)$ (i.e., $a \sim \mu(\cdot \mid s)$ when $s \sim \lambda$).

By definition, $v_u$ is the unique fixed point of $T_u : v \mapsto P\mu(\gamma v - u)$ on functions $v : \mathcal{S} \times \mathcal{A} \to \mathbb{R}$, so

$$v_u = P\mu(\gamma v_u - u) \quad \Longleftrightarrow \quad (I - \gamma P\mu)\, v_u = -(P\mu)u.$$

Hence

$$v_u = -(I - \gamma P\mu)^{-1}(P\mu)u, \qquad v_{u'} = -(I - \gamma P\mu)^{-1}(P\mu)u'.$$

Since $\gamma < 1$ and $P\mu$ is a contraction on $L^2(\lambda \otimes \mu)$ (by Jensen's inequality and stationarity of $\lambda \otimes \mu$),

$$\|(I - \gamma P\mu)^{-1}\|_{2,\lambda\otimes\mu} \le (1-\gamma)^{-1}.$$

Therefore,

$$\begin{aligned}
\|v_u - v_{u'}\|_{2,\lambda\otimes\mu} &= \left\|(I - \gamma P\mu)^{-1}\,(P\mu)(u' - u)\right\|_{2,\lambda\otimes\mu} \\
&\le \|(I - \gamma P\mu)^{-1}\|_{2,\lambda\otimes\mu} \, \|(P\mu)(u' - u)\|_{2,\lambda\otimes\mu} \\
&\le (1-\gamma)^{-1} \, \|u - u'\|_{2,\lambda\otimes\mu},
\end{aligned}$$

where the last inequality uses that $\|P\mu\|_{2,\lambda\otimes\mu} \le 1$. Thus $u \mapsto v_u$ is $(1-\gamma)^{-1}$-Lipschitz from $L^2(\lambda \otimes \mu)$ to $L^2(\lambda \otimes \mu)$. $\qquad\square$

*Proof of Lemma 4.* Let $e_k := \|v_{\hat u} - \hat v^{(k)}\|_{2,\lambda\otimes\mu}$. Since $v_{\hat u} = T_{\hat u} v_{\hat u}$ and $T_{\hat u}$ is a $\gamma$-contraction in $v$,

$$e_k \le \|T_{\hat u} v_{\hat u} - T_{\hat u} \hat v^{(k-1)}\|_{2,\lambda\otimes\mu} + \|\hat v^{(k)} - T_{\hat u} \hat v^{(k-1)}\|_{2,\lambda\otimes\mu} \le \gamma\, e_{k-1} + \eta_k.$$

Unrolling this recursion gives

$$e_K \le \gamma^K e_0 + \sum_{t=1}^K \gamma^{K-t}\eta_t,$$

which is the desired result. $\qquad\square$

*Proof of Theorem 3.* By the triangle inequality,

$$\|\hat v^{(K)} - v^\star\|_{2,\lambda\otimes\mu} \le \|\hat v^{(K)} - v_{\hat u}\|_{2,\lambda\otimes\mu} + \|v_{\hat u} - v^\star\|_{2,\lambda\otimes\mu}.$$

Apply Lemma 4 to the first term and Lemma 3 with $u' = u^\star$ to the second to obtain

$$\|\hat v^{(K)} - v^\star\|_{2,\lambda\otimes\mu} \le \gamma^K \|v_{\hat u} - \hat v^{(0)}\|_{2,\lambda\otimes\mu} + \sum_{t=1}^K \gamma^{K-t}\eta_t + \frac{1}{1-\gamma}\,\|\hat u - u^\star\|_{2,\lambda\otimes\mu}.$$

The specialization with $\eta_t \le \bar\eta$ follows by summing the geometric series. $\qquad\square$

*Proof of Lemma 6.* Let $u = \log \pi$, $\hat u = \log \hat\pi$, and define the likelihood ratio $\vartheta(a \mid s) := \hat\pi(a \mid s)/\pi(a \mid s)$. Since $\hat u, u \in \mathcal{U}$ are uniformly bounded from below and above, the log–likelihood ratio is also bounded $|\log \vartheta| \le L$ for some $L = 2B$. The function $\phi(t) := -\log t - (t-1)$ is $e^{-2L}$–strongly convex on $[e^{-L}, e^L]$, hence for each $s$,

$$\mathrm{KL}\big(\pi(\cdot \mid s) \,\|\, \hat\pi(\cdot \mid s)\big) = \mathbb{E}_\pi[-\log \vartheta] = \mathbb{E}_\pi[\phi(\vartheta)] \ge \frac{e^{-2L}}{2}\,\mathbb{E}_\pi\big[(\vartheta - 1)^2\big].$$

Moreover, by the mean value theorem on $[e^{-L}, e^L]$, $|\log \vartheta| \le e^L\,|\vartheta - 1|$, so

$$\mathbb{E}_\pi\big[(\hat u - u)^2\big] = \mathbb{E}_\pi\big[(\log \vartheta)^2\big] \le e^{2L}\,\mathbb{E}_\pi\big[(\vartheta - 1)^2\big] \le 2\,e^{4L}\,\mathrm{KL}\big(\pi(\cdot \mid s) \,\|\, \hat\pi(\cdot \mid s)\big).$$

Averaging over $s$ and taking square roots yields

$$\|\hat u - u\|_2 \le \sqrt{2}\,e^{2L}\left(\mathbb{E}_s \mathrm{KL}(\pi\|\hat\pi)\right)^{1/2}.$$

By Assumption 1 (KL generalization), $\mathbb{E}_s \mathrm{KL}(\pi\|\hat\pi) \le \hat\rho_{\mathcal{U}}^2(n/2, \delta)$ with prob. $\ge 1 - \delta$, hence

$$\|\hat u - u\|_2 \le C(L)\,\hat\rho_{\mathcal{U}}\big(\tfrac{n}{2}, \delta\big), \qquad C(L) := \sqrt{2}\,e^{2L}.$$

We turn to the second bound. As shorthand, let

$$(T_{\hat{u}}v)(s,a) := (P\mu)(\gamma v - \hat{u})(s,a) = \int \big[\gamma v(s',a') - \hat{u}(s',a')\big]\,\mu(da' \mid s')\,P(ds' \mid s,a).$$

By Assumption 2, $T_{\hat{u}}\hat{v}^{(k-1)} \in \mathcal{V}$ and is given by the population risk minimizer

$$T_{\hat{u}}\hat{v}^{(k-1)} \;=\; \arg\min_{v\in\mathcal{V}}\; \mathbb{E}_{(s,a)\sim P_{\text{data}}}\Big[(T_{\hat{u}}\hat{v}^{(k-1)})(s,a) - v(s,a)\Big]^2$$

$$=\; \arg\min_{v\in\mathcal{V}}\; \mathbb{E}_{(s,a,s')\sim P_{\text{data}}}\Bigg[\int\big(\gamma\hat{v}^{(k-1)}(s',a') - \hat{u}(s',a')\big)\,\mu(da' \mid s') - v(s,a)\Bigg]^2.$$

The estimator $\hat{v}^{(k)}$ is precisely the empirical risk minimizer over $v \in \mathcal{V}$ for the empirical least-squares analogue of the right-hand side. By Corollary 1 in Appendix C, for any $\delta \in (0,1)$, with probability at least $1 - \frac{\delta}{2K}$,

$$\|\hat{v}^{(k)} - T_{\hat{u}}\hat{v}^{(k-1)}\|_2 \;\lesssim\; \rho_{\mathcal{V}}\Big(m_K, \tfrac{\delta}{2K}\Big),$$

where hidden constants only depend on the function-class bound $B$. We apply the corollary conditional on the training data used to form the pseudo-outcome

$$Y(s,a,s') = \int \big(\gamma\hat{v}^{(k-1)}(s',a') - \hat{u}(s',a')\big)\,\mu(da' \mid s').$$

By sample splitting, the data used to fit $\hat{v}^{(k)}$ are conditionally i.i.d., and the LS-ERM assumptions (bounded outcomes, convex $\mathcal{V}$) hold. Moreover, $Y(s,a,s')$ is uniformly bounded since $\hat{u}$ and $\hat{v}^{(k-1)}$ are uniformly bounded by assumption on the classes $\mathcal{U}$ and $\mathcal{V}$.

The result now follows by collecting all bounds and noting that a union bound shows all events occur with probability at least

$$1 - \Big(\tfrac{\delta}{2} \;+\; K\cdot\tfrac{\delta}{2K}\Big) \;=\; 1 - \delta.$$

$\square$

*Proof of Theorem 4.* Intersect the events in Lemma 6 to obtain, with probability at least $1 - \delta$, the bounds $\nu := \|\hat{u} - u^\star\|_2 \le \hat{\rho}_{\mathcal{U}}(n/2, \delta/2)$ and $\eta_k := \|\hat{v}^{(k)} - A_{\hat{u}}\hat{v}^{(k-1)}\|_2 \le \hat{\rho}_{\mathcal{V}}(m_K, \delta/(2K))$ for all $k$. Plug these into Theorem 3 to get

$$\|\hat{v}^{(K)} - c^\star\|_2 \;\le\; \frac{1}{1-\gamma}\,\hat{\rho}_{\mathcal{U}}\big(\tfrac{n}{2}, \tfrac{\delta}{2}\big) \;+\; \gamma^K\|v_{\hat{u}} - \hat{v}^{(0)}\|_2 \;+\; \sum_{t=1}^{K}\gamma^{K-t}\,\hat{\rho}_{\mathcal{V}}\big(m_K, \tfrac{\delta}{2K}\big),$$

and summing the geometric series yields the displayed inequality. The simplified bound for $K \simeq c\log n$ follows by noting that $\gamma^K$ decays polynomially in $n$ and is dominated by the statistical terms. $\square$

*Proof of Lemma 5.*

$$\hat{r}^{(K)} - r^\star = (\hat{u} - u) + \mu\big[\gamma(\hat{v}^{(K)} - v^\star) - (\hat{u} - u)\big] - \gamma\,(\hat{v}^{(K)} - v^\star) = (\mu - I)\big[\gamma(\hat{v}^{(K)} - v^\star) - (\hat{u} - u)\big].$$

Now add–subtract a Bellman step. Let

$$\varepsilon_K \;:=\; \hat{v}^{(K)} - T_{\hat{u}}\hat{v}^{(K-1)}, \qquad T_{\hat{u}}v \;:=\; (P\mu)\big(\gamma v - \hat{u}\big).$$

Then

$$\hat{r}^{(K)} - r^\star = (\mu - I)\Big(\gamma\big(T_{\hat{u}}\hat{v}^{(K-1)} - v^\star\big) - (\hat{u} - u)\Big) \;+\; (\mu - I)\big[\gamma\,\varepsilon_K\big].$$

Thus

$$\hat{r}^{(K)} - r^\star = (\mu - I)\Big[-(I - \gamma P\mu)(\hat{u} - u) \;+\; \gamma^2(P\mu)(\hat{v}^{(K-1)} - v^\star)\Big] \;+\; \gamma(\mu - I)\,\varepsilon_K.$$

Assume the coverage condition

$$\kappa := \operatorname*{ess\,sup}_{(s,a)} \frac{dP}{d(\lambda\otimes\mu)}(s,a) < \infty$$

and stationarity $\lambda P\mu = \lambda$. Then, by Jensen and stationarity,

$$\|(P\mu)f\|_{2,\lambda\otimes\mu} \leq \|f\|_{2,\lambda\otimes\mu},$$

and by change of measure,

$$\|(P\mu)f\|_2 \leq \kappa^{1/2}\|(P\mu)f\|_{2,\lambda\otimes\mu} \leq \kappa^{1/2}\|f\|_{2,\lambda\otimes\mu}.$$

Hence,

$$\|\hat{r}^{(K)} - r^\star\|_2 \;\lesssim\; \|\hat{u} - u\|_2 + \|(P\mu)(\hat{u} - u)\|_2 + \|(P\mu)(\hat{v}^{(K-1)} - v^\star)\|_2 + \|\varepsilon_K\|_2,$$

and using the bounds above,

$$\|\hat{r}^{(K)} - r^\star\|_2 \;\lesssim\; \|\hat{u} - u\|_2 + \|\hat{v}^{(K-1)} - v^\star\|_{2,\lambda\otimes\mu} + \|\varepsilon_K\|_2,$$

with all hidden constants depending only on $\kappa$ and $\gamma$.

$\square$

## C  HIGH-PROBABILITY BOUND FOR LEAST SQUARES

The following theorem is, up to notation, equivalent to Theorem 5.2. in Koltchinskii (2011)

**Theorem 5** (Theorem 5.2. in Koltchinskii (2011)). *Let $\mathcal{G}$ be a convex class of bounded functions and let $\hat{g}$ denote the least squares estimator of the regression function*

$$\hat{g} \;:=\; \arg\min_{g\in\mathcal{G}} \frac{1}{n}\sum_{j=1}^{n}(Y_j - g(X_j))^2,$$

*where each $Y_j$ is almost surely uniformly bounded.*

*Then, there exist constants $K > 0$, $C > 0$ such that for all $t > 0$,*

$$\mathbb{P}\left\{ \|\hat{g} - g^\star\|_2^2 \;\geq\; \inf_{g\in\mathcal{G}} \|g - g^\star\|_2^2 \;+\; K\left(\hat{r}_{\mathcal{G}}(n)^2 + \tfrac{t}{n}\right)\right\} \;\leq\; Ce^{-t},$$

*where*

$$\hat{r}_{\mathcal{G}}(n) \;:=\; \inf\left\{ r > 0 : \hat{\mathfrak{R}}_n(\mathcal{G}, r) \lesssim r^2\right\}, \qquad \hat{\mathfrak{R}}_n(\mathcal{G}, r) := \mathbb{E}_\varepsilon\left[ \sup_{g,h\in\mathcal{G}:\,\|g-h\|_2\leq r} \frac{1}{n}\sum_{i=1}^{n}\varepsilon_i\{g - h\}(X_i)\right].$$

We have the following corollary.

**Corollary 1** (PAC form; well-specified). *Under the conditions of Theorem 5 and assuming $g^\star \in \mathcal{G}$, for any $\delta \in (0, 1)$, with probability at least $1 - \delta$,*

$$\|\hat{g} - g^\star\|_2^2 \;\leq\; K\left(\hat{r}_{\mathcal{G}}(n)^2 + \tfrac{1}{n}\log\tfrac{1}{\delta}\right).$$

*Proof.* By Theorem 5, for all $t > 0$,

$$\Pr\left\{ \|\hat{g} - g^\star\|_2^2 \;\geq\; \inf_{g\in\mathcal{G}} \|g - g^\star\|_2^2 + K\left(\hat{r}_{\mathcal{G}}(n)^2 + \tfrac{t}{n}\right)\right\} \leq Ce^{-t}.$$

Under $g^\star \in \mathcal{G}$, the infimum is 0. Set $t = \log(C/\delta)$ so that $Ce^{-t} = \delta$. Then with probability at least $1 - \delta$,

$$\|\hat{g} - g^\star\|_2^2 \;\leq\; K\left(\hat{r}_{\mathcal{G}}(n)^2 + \tfrac{1}{n}\log\tfrac{C}{\delta}\right) \;\leq\; K'\left(\hat{r}_{\mathcal{G}}(n)^2 + \tfrac{1}{n}\log\tfrac{1}{\delta}\right),$$

absorbing $\log C$ into $K'$. $\square$

## D  USE OF LARGE LANGUAGE MODELS

ChatGPT 5 was used to provide an initial skeleton of the paper, given a prompt of what the paper is about and the main technical ideas. This skeleton, which had messy notation, inaccurate theorems, and incorrect references, was then iterated on extensively by the human authors to achieve the submitted paper. We nonetheless found the initialization very helpful to jump start the writing process.

