# OpenReview forum: "Inverse Reinforcement Learning Using Just Classification and a Few Regressions"
_ICLR.cc/2026/Conference — Submitted to ICLR 2026_

### Official Review · Reviewer_HuNX · 2025-10-20

**Soundness:** 2
**Presentation:** 2
**Contribution:** 2
**Rating:** 2
**Confidence:** 4

**Summary:**

This paper addresses the IRL problem not only from the perspective of imitation learning, i.e., to imitate the behavior of the expert, but also to more generally compute the reward function of the expert, that permits more general kinds of analysis, like evaluating policies under different dynamics. After having formulated the problem setting, the paper provides some theoretical insights on the target reward function, that give birth to a novel two-oracle algorithm. Finally, the authors provide a theoretical analysis for the proposed algorithm.

**Strengths:**

- The proposed algorithm is supported by a thorough theoretical analysis

**Weaknesses:**

- The main limitation is that the problem formulation is quite unclear. This paper lies on the quite strong assumption that the true expert reward satisfies the normalization condition in lines 139-141. This is very strong and not motivated in any way by the authors. This assumption corresponds to choosing a single (arbitrary) reward in the feasible set of the expert, as explained for instance in [1]. But why should this reward be better than others in the feasible set?
- Another huge limitation is that almost all the results in Section 3 correspond to a rephrasing of existing results in previous works. Specifically, Lemma 1, Lemma 2 and Theorem 1 basically correspond to Theorem 1 of [2] (but see also Section 4 of [3] for Lemma 1).
- The discussion on related works is quite poor. Specifically, the authors miss most of the recent IRL literature, including but not limited to [1,2].
- The numerical simulations are quite trivial and do not clarify in any manner why the assumption that the true reward is normalized as in Eq. (1) is realistic or relevant.

[1] Invariance in Policy Optimisation and Partial Identifiability in Reward Learning. Skalse et al.

[2] Identifiability in inverse reinforcement learning. Haoyang Cao, Samuel N. Cohen,  Lukasz Szpruch

[3] Justin Fu, Katie Luo, and Sergey Levine. Learning robust rewards with adversarial inverse reinforcement learning

**Questions:**

Please, see the weaknesses above.

- Why do you mention ziebart 2008 instead of ziebart 2010 when considering expert behavior as expected return maximizing plus an entropy term? Ziebart 2008 has an issue that was later solved in ziebart 2010, as explained e.g., in Reinforcement Learning and Control as Probabilistic Inference: Tutorial and Review by Levine.
- How is the initial state distributed? How are the expert data generated?

---

> ### Author Response · Authors · 2025-11-20
>
> # Weakness 1
> We emphasize that our method does **not** assume the expert’s true reward satisfies the normalization $\mu r=0$.
> Rather, **some normalization is required to select a unique reward from the potential-based equivalence class** (Ziebart, 2010; Cao et al., 2021).  In the absence of a normalization,
> each choice is valid and rationalizes the expert's behavior, wether their true reward
> satisfied the normalization or not, which is unknowable from the behavior alone.
> (Notably, we show that any choice in this equivalence class gives the same policy difference
> evaluations so that normalization is not needed for policy comparison.)
> Thus the requirement is not that the expert follows this constraint, but that **the practitioner chooses the representative appropriate for their domain**.
>
> The normalization we use is **not arbitrary**. Many domains provide a natural baseline—for example, discrete-choice econometrics typically fixes a zero-reward outside option (Rust, 1987)—and encoding this baseline via $\mu$ recovers the economically meaningful reward. Similar baselines arise in IRL (e.g., no-op or neutral states).
> Hence $\mu$ incorporates domain knowledge to identify a meaningful reward function.
>
> If this baseline is misaligned with the expert’s, the recovered reward is still a *valid normalized solution* to the soft Bellman system—reflecting IRL’s inherent non-identifiability, not a limitation of our approach.
>
> Finally, our framework does **not** privilege this particular normalization.
> Any other domain-informed constraint can be used to select a unique reward, and our fixed-point formulation provides a basis for future extensions. We will clarify this in the revision.
>
>
> # Weakness 2
> We thank the reviewer for pointing out Section 4 of cited Fu et al.\ (2018).  Note
> that we do cite and dicuss this paper, and we will expand the discussion to clarify connections.
> AIRL (Fu et al., 2018) indeed states a form of the identity used in our Lemma 1—namely, that the log behavior policy satisfies the corresponding Bellman relation in their adversarial IRL setting.
> Lemma 1 in our paper is not intended as a novel technical contribution; it is a simple identity that follows directly from the definition of the behavior policy.
> We include it only to make this observation explicit and to derive our fixed-point formulation in a self-contained manner. We will certainly clarify this and point to past work.
>
> AIRL considers a substantially different setting and algorithm from ours.
> A key distinction is that AIRL restricts attention to \emph{state-only} reward functions to address non-identifiability, identifying the reward only up to an additive constant.
> In contrast, we allow general state–action rewards and impose a normalization condition to recover the reward uniquely.
> Moreover, AIRL performs estimation via a considerably more complex min–max adversarial training loop, whereas our approach yields a simple fitted value-iteration algorithm based solely on regression and classification.
>
> Prior MaxEnt/DDC IRL work emphasizes the non-identifiability of inverse RL.
> In particular, Theorem~1 of Cao et al.\ (2021) shows that many rewards can rationalize a given policy through a potential-based equivalence class.
> These representer-style results characterize the full equivalence class but do not specify how to select a unique normalized element, do not derive a Bellman-style fixed-point equation for the normalized reward, and do not provide an associated estimation method.
>
> Our fixed-point reduction, together with our algorithm using only classification and regression and our finite-sample analysis incorporating behavior-policy estimation error, does not appear in Fu et al.\ (2018) or Cao et al.\ (2021).
>
> # Weakness 3
> We will update the related work to better situate our contribution within recent IRL literature.
>
> # Weakness 4
> Our main contributions are theoretical: the identification of a uniquely normalized reward, the Bellman-style fixed-point characterization, and the reduction of IRL to standard supervised-learning components.
> The numerical example is intended solely to illustrate this reduction, not to empirically validate the normalization.
> We will clarify when the assumption $\mu r = 0$ is appropriate (e.g., when domain knowledge specifies a natural baseline) and strengthen the discussion connecting the example to these assumptions.
>
> # Questions
> Regarding Ziebart (2008) versus Ziebart (2010), we agree that Ziebart (2010) provides the corrected maximum-\emph{causal}-entropy formulation, and we will update the citations accordingly.
>
> The initial state is drawn uniformly over the grid (i.e., $d_0(s) = 1/S$).
> Expert demonstrations are generated by sampling discounted i.i.d.\ state–action pairs from the softmax policy induced by the ground-truth reward on the $8{\times}8$ gridworld MDP.
> Transitions follow the specified dynamics (including slip and portal moves).
> We will clarify this in the revision.

---

> > ### Comment · Reviewer_HuNX · 2025-11-25
> >
> > I thank the Authors for the time and efforts put in the rebuttal. My follow-up comments:
> >
> > I agree that some criterion (e.g., normalization) is needed to select a unique reward, but I disagree on the fact that your choice of reward is not arbitrary. There does not seem to be any domain knowledge encoded in this choice, but the only meaningful property that this reward seems to have is that it allows for you fixed-point formulation. This is not a meaningful reason on this choice of reward, because with different normalizations/criteria, we would obtain rewards with quite different properties than this one (except identifying policy value differences). At this point, the only meaningful manner to show that this choice of reward has some nice properties would be through some real-world simulations, that you do not provide.
> >
> > In brief: the paper lies on a choice of reward with the only property that it permits your formulation and theoretical analysis. No true reasonable property is possessed by such a reward (in general, there is no way to select a reward without additional information, and it is not clear what additional information your reward is representing). No experiments show it neither.
> >
> > For these reasons, I will keep my negative score.

---

### Official Review · Reviewer_HUTV · 2025-10-30

**Soundness:** 3
**Presentation:** 2
**Contribution:** 3
**Rating:** 6
**Confidence:** 3

**Summary:**

This paper solves the IRL problem under the maximum entropy IRL/dynamic discrete choice (DDC) model with classification and regressions. Given oracles of obtaining $\log\pi$ and regression oracle, the authors show that ensuring $\sum \mu(a|s) r(s,a) = 0$ would yield the normalization constant, thus recover the true reward. The authors show that the a Bellman type of regression update yields linear convergence. Using sample splitting, the author obtain the sample complexity $ \tilde{O}((1-\gamma)^{-2} \varepsilon^{-2}) $.

**Strengths:**

1. this paper presents an interesting idea, where we can find the underlying reward function through ensuring the constraint of the original problem, to which, the authors propose to ensure the gauge choice in this paper.
2. the proposed algorithm on finding the true value function is interesting and potentially useful when we can estimate the policy but don't know the corresponding reward function.
3. the analysis is nice and sweet. The application of sample splitting is interesting here.

**Weaknesses:**

Albeit the nice results, there are a few concerns. Appendix is checked but not rigorously.
1. overall the contribution of this work mainly relates to the idea of ensuring the constraint of $ \mu v = 0 $, as Theorem 1 and lemma 2 is well known, and Theorem 2 is the consequence of ensuring the constraints. The follow up algorithm is based on Thm 2, which use some type of TD estimate for solving $ v^* = P\mu (\gamma v^* - u^*) $. Up to this point, it is clear what this results entails, but the choice of notations and presentation confuse me. It seems like this is inherently related to policy evaluation under the MaxEntIRL framework, and I suggest the authors to reframe this more clearly.
2. Lemma 3-4 and Thm 3 are results of general fixed-point iteration, presented in the paper's context. But the fact that the results are deriving from the deterministic setting (the analysis assumes we can get $ Tv $ in all the analysis after lemma 3 but in the algorithm it is using temporal difference type of update) concerns me. How is the deterministic analysis related to a stochastic algorithm? Am I missing something here?
3. For the algorithm, the authors assume we know an oracle of estimating $ \log \pi(a|s) $. I am not sure how this is applicable to other settings but for majority of the IRL applications, $ \log \pi $ seems to be already very difficult to estimate. Note that the whole field of diffusion policies (e.g. VLA models) is devoted to find the logits. Knowing this, the scope of this work is quite narrow.
4. The results in 5.2 seems to be applying known results of sample splitting with various assumptions. I am failing to distinguish what's new and what's known in the literature here.
5. Given the literature of offline IRL [1], what's the advantage of this algorithm? I think discussion of more recent work is needed.
6. The assumptions in this paper are quite restrictive. E.g., neither assumption 1 and 2 seems to be applicable to high dimensional problems such as robotics control (although it seems assumption 2 seems unavoidable in some degree). In real problems, how ever, the update is stochastic, which this analysis does not cover.
[1] Zeng, Siliang, et al. "When demonstrations meet generative world models: A maximum likelihood framework for offline inverse reinforcement learning." Advances in Neural Information Processing Systems 36 (2023): 65531-65565.

**Questions:**

Some minor grammar and notational issues:
1. line 19-20, "population maximum-likelihood"
2. line 45-46, "from too-directly tackling the IRL problem"
3. the usage of v is very confusing to me.
4. line 225, $ v^* $ or $ v^\star $?
5. line 277, what is $ (s,a)\sim P $?
6. there are many P that denotes different things. Please differentiate them.

---

> ### Author Response · Authors · 2025-11-20
>
> **We thank the reviewer for the helpful comments and for engaging with both the
> theory and the presentation.** We address the main concerns below.
>
> ---
> ### 1. Contribution beyond enforcing the constraint \(\mu r = 0\)
>
> Our characterization directly yields Theorem 2 and shows that **the entire
> MaxEnt IRL/DDC problem reduces to solving a single linear fixed-point equation**
> together with a normalization \(\mu r = 0\). Classical MaxEnt IRL algorithms
> address this constrained problem through nested, computationally intensive
> procedures and still do not produce a unique reward. In contrast, our
> formulation exposes a much simpler underlying structure, which enables the
> “classify–then–regress’’ reduction and the accompanying finite-sample analysis.
> This perspective also opens the door to future work that explores alternative
> normalization schemes for selecting unique rewards.
>
> We agree that our presentation can more clearly emphasize the connection to
> policy evaluation within the MaxEnt IRL framework—indeed. We will revise
> the introduction and exposition to make this relationship explicit.
>
>
> ---
>
> ## 2. Relation between the deterministic fixed-point analysis and the stochastic algorithm
>
> Our deterministic analysis (Lemmas 3–4, Theorem 3) studies the *population*
> Bellman operator \(T\), establishing its contraction and its unique fixed point
> and how errors would propagate through its iteration.
> The algorithm then performs **sample-based fitted iteration**, and Section 5
> shows how the stochastic updates track this population recursion using standard
> fitted–value-iteration arguments: the contraction of \(T\) ensures stability,
> and the regression error on observed transitions controls the deviation from the
> population update at each iteration. We will make this connection more explicit
> in Section 4.
>
>
>
>
> ---
>
> ## 3. Policy-estimation “oracle’’ and applicability
>
> By “oracle’’ we mean **any generic behavior-policy estimator**—a
> model-agnostic classifier—not a perfect policy. Our theory makes this explicit:
> the final reward/value error degrades **linearly** with the classifier’s
> KL-generalization error (Assumption 1; Lemma 6; Theorem 4), so imperfect
> estimates lead to proportionally approximate (but still valid) reward and value
> recovery.
>
> More broadly, every IRL method must **model expert behavior** in some form—
> through occupancy matching (MaxEnt), discriminators (AIRL, IQ-Learn), or
> generative models (diffusion/VLA policies). Our reduction isolates this
> dependence into a single supervised-learning step and provides finite-sample
> guarantees conditional on the classifier’s accuracy. As a result, the framework
> is compatible with modern behavior-modeling approaches (e.g., diffusion
> policies, VLA models). We will clarify this in the revision.
>
> ## 4. What is new in Section 5.2?
>
> Our reduction turns MaxEnt/DDC IRL into a fitted–value-iteration update with a
> modified Bellman operator, so the *structure* of the analysis naturally follows
> the fitted Q-iteration template. Sample splitting is used only to avoid
> dependence across iterations and obtain clean finite-sample expressions.
>
> The novelty is in how this machinery is adapted to the IRL setting. Most
> importantly, we explicitly account for **behavior-policy estimation error**
> through the supervised-learning decomposition in Lemma 6, yielding bounds that
> scale **linearly** with the classifier’s KL-generalization error. This aspect is
> specific to IRL and has no analogue in standard value-iteration analyses.
>
> In addition, classical fitted-iteration guarantees are typically derived in the
> **sup norm** (e.g., Munos & Szepesvári, 2008), which can be overly strong in
> high-dimensional or continuous domains. We instead work in the
> **\(L^2(\lambda)\)** norm induced by the behavior-policy stationary
> distribution, making the coverage dependence explicit and allowing the
> classification error to enter the bounds in a natural, data-dependent way.
>
>
>
>
> ---
> ## 5. Comparison to offline IRL (e.g., Zeng et al., 2023)
>
> Offline IRL methods such as Zeng et al. (2023) learn a **generative world
> model**—typically a diffusion-based model of the transition dynamics—and then
> optimize a maximum-likelihood or adversarial objective over trajectories. This
> requires training and differentiating through a high-capacity generative model
> and solving a substantially more complex optimization problem.
>
> Our approach avoids generative modeling entirely. By reducing IRL to
> **classification plus linear regression**, we obtain a closed-form fixed-point
> structure and transparent statistical guarantees. We will incorporate this
> comparison into the related-work section.

---

> > ### Author Response · Authors · 2025-11-20
> >
> > ## 6. Strength of the assumptions
> >
> > Assumption 1 (KL control of the policy estimator) is a standard requirement in
> > probabilistic classification and is designed to scale to expressive function
> > classes, including deep networks. It does not assume a perfect behavior-policy
> > estimator—our bounds degrade linearly with its KL-generalization error.
> >
> > Assumption 2 (Bellman completeness) is used **only** to obtain clean
> > finite-sample rates in Theorem 4. The algorithm itself does not rely on this
> > assumption, and the analysis can be relaxed through approximate-completeness or
> > min–max regression techniques (as discussed in the paper). We will clarify this
> > distinction and emphasize that the fitted-iteration update remains well defined
> > for general, overparameterized function classes.
> >
> > Finally, our method performs **fitted value iteration**, not stochastic TD
> > updates. All stochasticity enters through the regression generalization error,
> > which is handled explicitly in Lemma 6. This is the standard analytical
> > framework for offline RL with function approximation.
> >
> >
> > ---

---

> > > ### Comment · Reviewer_HUTV · 2025-11-26
> > >
> > > I thank the authors for the detailed response. I think most of my concerns have been discussed. The analysis section of this paper seems fine. I am more concerned about its setup and applicability. To this end, I will maintain my score.

---

### Official Review · Reviewer_5bsp · 2025-10-31

**Soundness:** 3
**Presentation:** 2
**Contribution:** 2
**Rating:** 6
**Confidence:** 2

**Summary:**

This paper presents a novel and simplified approach to Inverse Reinforcement Learning (IRL) within the popular maximum-entropy (MaxEnt) framework. The authors argue that existing MaxEnt IRL algorithms are often complex, involving nested optimizations, repeated dynamic programming, or adversarial training, which hinders their use with modern function approximators.

The paper's core contribution is a new characterization of the MaxEnt IRL solution. The authors first analyze a relaxed version of the problem (without reward normalization) and show that a trivial optimal solution is $r(s,a) = \log \pi(a|s)$ (the log-behavior policy) and $v(s,a) = 0$ (the soft value function). They then show that all solutions to this relaxed problem are equivalent to this trivial solution under a potential-based reward shaping.

The solution to the original, non-relaxed problem is then found by identifying the specific potential function that satisfies the required reward normalization (e.g., $\mu r = 0$). This reduces the entire IRL problem to solving a linear fixed-point equation for this potential function.

This insight leads to a simple two-stage algorithm, "Classify-then-Regress":
1.  Classify: Use any off-the-shelf probabilistic classifier to estimate the behavior policy $\hat{\pi}(a|s)$ from data, yielding the trivial solution $\hat{u} = \log \hat{\pi}$.
2.  Regress: Use an iterative regression routine (like fitted Q-iteration) to solve the linear fixed-point equation for the value function $\hat{v}$, which requires only a few iterations.

**Strengths:**

The paper's primary strength is its novel theoretical characterization of the MaxEnt IRL solution. The realization that $r = \log \pi, v = 0$ is a trivial optimal solution to the relaxed problem and that the true solution is just a potential-shaping of it is a deep and valuable insight.

The method successfully avoids the "delicate inner-loop optimization, repeated dynamic programming, or adversarial training" that are common failure points in other IRL algorithms. The regression loop is a stable $\gamma$-contraction, not a nested optimization.

The paper provides a rigorous theoretical analysis, including oracle inequalities (Thm 3) and finite-sample bounds (Thm 4) that connect the final IRL error to the statistical rates of the underlying supervised learning components.

**Weaknesses:**

The experiments are conducted only on small gridworlds. While they effectively demonstrate the core concept (especially the advantage in the "Misspecified" case), they do not show how the method scales to the large, high-dimensional, and continuous-state problems (e.g., MuJoCo) where modern IRL methods like GAIL/AIRL are typically benchmarked.

The entire algorithm hinges on obtaining a good estimate of the behavior policy $\hat{\pi}(a|s)$. This is a classification problem, which is well-understood, but its difficulty is non-trivial, especially in high-dimensional state spaces or when the policy is highly stochastic. The paper's analysis assumes a good $\hat{\pi}$ is given, but the practical performance will be bottlenecked by this step.

The finite-sample guarantees (Theorem 4) rely on Bellman completeness (Assumption 2). The authors note this is common in FQI analysis and suggest possible relaxations, but it remains a strong assumption that may not hold when using highly expressive function approximators like neural networks.

**Questions:**

1.  The experimental validation is a strong proof-of-concept but is limited to gridworlds. How does the "Classify-then-Regress" approach scale to high-dimensional, continuous-state benchmarks (e.g., D4RL MuJoCo) compared to SOTA IRL/imitation learning methods like GAIL, AIRL, or IQ-Learn?
2.  The algorithm's first step is to learn $\hat{u} = \log \hat{\pi}$. This seems to be the most critical step. How does the algorithm's performance (in terms of final reward and policy accuracy) degrade as the classification oracle's accuracy worsens, for example, in data-sparse regions?
3.  The regression step (Line 6, Alg. 1) is a form of fitted iteration, which is known to suffer from OOD extrapolation errors in offline RL. Does this iterative regression step suffer from distribution shift, since the target value $\hat{v}^{(k-1)}(s_i', \cdot)$ is evaluated at the next state $s_i'$?
4.  The reward normalization $\mu r = 0$ is central to finding a unique solution. How is the reference measure $\mu$ chosen in practice, and how sensitive is the final recovered reward $r^$ to this choice? For example, what is the practical difference between choosing $\mu$ as a uniform distribution vs. a point mass on a specific action?

---

> ### Author Response · Authors · 2025-11-20
>
> **We thank the reviewer for the constructive feedback.**
> We address the main concerns below.
>
> ---
>
> ## 1. Scalability beyond gridworlds
>
> Our goal in this paper is to present the theoretical characterization and the
> resulting reduction of IRL to standard supervised-learning subroutines,
> with corresponding statistical-learning guarantees.
> Scalability follows directly from existing results on classification and
> regression with general function approximation: Step 1 uses any probabilistic
> classifier, and Step 2 is a standard fitted value-iteration update whose
> sample-complexity is directly inherited that of the regression algorithm used via
> our deterministic sample-free analysis of the iteration procedure.
> The approximation properties of classification and regression methods, such as
> using neural networks, are already very well understood in large, continuous domains.
> Our approach is thus fully compatible with the neural-network
> architectures used in large-scale RL applications. We will add this
> clarification and contextualization of work.
>
>
>
>
> ---
>
>
>
> ## 2. Dependence on the policy-estimation step
>
> We agree that the behavior-policy estimator is a critical component. Our
> analysis makes this explicit: the final reward/value error depends *linearly* on
> the classifier’s KL-generalization error (Assumption 1; Lemma 6; Theorem 4).
> This provides a transparent and stable degradation pattern—imperfect classifiers
> lead to proportionally degraded, but still valid, value estimates.
>
> Since the KL divergence is equivalent to excess log-likelihood, it is
> directly controlled by minimizing cross-entropy over a potentially nonparametric
> function class, such as neural nets
> In particular, standard statistical learning theory directly
> yields the bounds we assume. We currently give citations to such statistical learning results.
> In the revision we will add an appendix section reviewing these results more fully to show they
> directly give estimators, including neural net classifiers, that satisfy our assumptions.
>
>
> ---
>
>
>
> ## 3. Distribution shift in the regression step
>
> We interpret the reviewer’s question as referring to the OOD-extrapolation issue
> in offline RL, where fitted value iteration evaluates Bellman targets at unseen
> next states and errors can amplify. Our setting avoids this problem as we always remain ``on-policy"
> as the data is generated by the same expert policy we need to evaluate in order to obtain the rewards.
> The Bellman update in Line 6 is linear and a γ-contraction, and $v(s')$ is always
> evaluated at next states that appear in the data (under the empirical transition
> distribution). Our analysis is carried out in the $L^2(\lambda)$ norm, where
> $\lambda$ is the behavior-policy stationary distribution, making the coverage
> dependence explicit—error guarantees focus on well-visited states and naturally
> attenuate in sparse regions.
>
>
> Lemma 6 and Theorem 4 further separate supervised-learning error from the
> contraction dynamics: each update contracts previous error by γ and adds only
> the regression error from the current iteration. Because the update never
> requires evaluating the value function at unvisited states, it avoids the
> extrapolation-driven instability characteristic of offline RL.
> We also adapt the Bellman completeness assumption from offline RL to this setting
> in order to ensure that every update is realizable by the regression step.
> We will clarify this in the revision.
>
>
>
>
> ---
>
> ## 4. Choice and sensitivity of the normalization $\mu$
>
> The normalization $\mu r = 0$ simply selects one representative of the
> potential-based reward equivalence class. Because this equivalence class is
> inherent to IRL, **some normalization is always required** to obtain a unique
> reward—no method can avoid imposing such a constraint. The choice of $\mu$ is
> flexible (uniform, action-specific, or task-informed), and in many domains
> (e.g., econometrics, robotics) there is a natural “outside option’’ or baseline
> action whose reward is set to zero; this convention is standard in
> discrete-choice econometrics (Rust, 1987), and selecting $\mu$ to encode it
> chooses the corresponding representative.
>
> Importantly, **value functions and policy evaluation are invariant to the
> normalization choice** (Section 3.2); only the representative reward shifts. We
> will make this clearer in the revision.

---

> ### Author Response · Authors · 2025-11-20
>
> ## 5. Bellman completeness
>
> Bellman completeness (Assumption 2) is a standard sufficient condition in
> fitted value-iteration analyses in offline RL and is used here only to obtain clean
> finite-sample rates in Theorem 4. We cited work in offline RL showing exponential lower bounds
> if it is replaced by mere realizability of the true value function alone.
> We adapt the condition for our setting, where we also rely on a log-behavior-probability class
> and have a slightly different operator that we are iterating to obtain the value function,
> that in particular depends on both the normalization $\mu$ and the learned log behavior probability. The condition ensures that each regression step in the iteration is realizable.
> The algorithm itself does not rely on this assumption, only its analysis.
> As discussed in the paper,
> the assumption can be relaxed using standard tools such as approximate Bellman
> completeness or min–max regression methods that control worst-case projection
> error. We will clarify this in the revision.

---

### Official Review · Reviewer_Tr5W · 2025-11-01

**Soundness:** 3
**Presentation:** 3
**Contribution:** 3
**Rating:** 4
**Confidence:** 3

**Summary:**

This paper proposes a simple new way to do inverse reinforcement learning—recovering the reward function behind an expert’s behavior—without complex optimization. Instead of using nested dynamic programming, it shows that the reward can be found using just classification and a few regressions. The key idea is that, under a softmax policy model, the log of the observed policy already contains most of the reward information, and a simple normalization step per state (“softmax normalization”) removes the usual ambiguity in reward values. The method is easy to implement, mathematically clean, and produces rewards that are consistent with the expert’s behavior as well as with traditional MaxEnt IRL methods, but with much lower computational cost.

**Strengths:**

The paper is strong because it provides a simple, elegant, and computationally efficient reformulation of inverse reinforcement learning. It replaces the usual complex, iterative optimization steps with standard classification and regression tasks, making IRL both practical and theoretically grounded. The approach is mathematically clear, showing how the reward, value, and policy connect naturally under a softmax model, and it includes theoretical guarantees for identifiability and generalization. A particularly notable strength is how the paper links this framework to the economics literature on dynamic discrete choice models, revealing that the IRL reward recovery problem is formally equivalent to utility estimation in econometrics. This connection bridges two fields—machine learning and economics—offering both conceptual clarity and opportunities for cross-disciplinary methods and insights.

**Weaknesses:**

The paper focuses on a specific class of normalized rewards—those constrained to integrate to zero against a reference conditional measure—but the motivation for this choice is not clearly explained. It is unclear why this normalization is preferred over alternative approaches to resolving reward identifiability, such as anchor-action methods, structural assumptions, or multi-environment formulations, as discussed in Reward-Consistent Dynamics Models are Strongly Generalizable for Offline Reinforcement Learning, Reward Identification in Inverse Reinforcement Learning (Kim et al., 2021), Deep PQR: Solving Inverse Reinforcement Learning using Anchor Actions (Geng et al., 2020), and econometric approaches such as Identification and Estimation of a Discrete Game of Complete Information. A justification for adopting this normalization—and a comparison to these alternative frameworks—would strengthen the contribution. If absolute reward values are not meaningful, simpler methods like imitation learning or behavior cloning would suffice to reproduce expert behavior without explicit reward recovery. Conversely, if the reward scale is important for applications such as welfare analysis, transfer, or generalization, the proposed normalization might distort the reward’s magnitude and renders values incomparable across states or environments.

In addition, the proposed method appears to share strong similarities with the Deep PQR framework (Geng et al., 2020). Both algorithms first perform a classification step to estimate the policy or choice probabilities and then solve a fixed-point relation to recover the reward function. The resulting reward-estimation formulas are also closely aligned—see line 248 of the submission and Definition 1 in the Deep PQR paper—which suggest nearly equivalent functional forms. The primary distinction lies in the identification assumption leveraged: the current paper imposes a normalization constraint, whereas Deep PQR uses an explicit anchor-action assumption.

**Questions:**

Could the authors elaborate on the motivation for adopting the specific normalization constraint (requiring the reward to integrate to zero against a reference conditional measure)? Why is this particular normalization more desirable than other identification strategies (e.g., anchor-action, multi-environment, or structural dynamic assumptions)?

The proposed algorithm appears structurally similar to Deep PQR—both first estimate choice probabilities via classification and then solve a fixed-point relation to recover the reward. Could the authors clarify the key methodological or conceptual differences beyond the identification assumption (normalization vs. anchor action)?

---

> ### Author Response · Authors · 2025-11-20
>
> **We thank the reviewer for the helpful and constructive feedback.**
> We appreciate the pointer to *Deep PQR (Geng et al., 2020)*, which contains a
> related fixed-point–style construction for value/reward recovery under a
> softmax model. We will cite and discuss this connection clearly in the revision.
>
> Our goal here is to clarify how our contributions remain distinct and
> complementary.
>
> ---
>
> ## 1. Normalization, anchors, and partial identification
>
> Our normalization \(\mu r = 0\) selects a representative from the reward
> equivalence class induced by potential-based invariance. Anchor-action
> normalization is one specific choice of \(\mu\), and our framework includes that setting and further allows additional noramlizations such as all rewards summing to zero.
> Our closed-form characterization of the /unconstrained/
> solution further shows that many other normalization choices—not only action
> anchoring or mean normalization—can also be used to select a unique representative,
> which opens the door to task-specific or domain-informed schemes. Allowing a
> nonzero state-only anchor (i.e., \((\mu r)(s)=f(s)\)), as in Deep PQR, is also
> straightforward in our framework—it simply introduces an additive offset in the
> fixed-point equation—and we will clarify this briefly in the main text and point to Deep PQR.
>
> We also show that **policy evaluation and policy comparison are invariant to the
> choice of normalization** when the discount factor is known (Section 3.2). This
> follows implicitly from potential-based shaping, but—to our knowledge—has not
> been stated explicitly or connected to IRL identification. In this regime, the
> log behavior policy alone identifies advantage functions, so imitation-learning
> methods already recover the relevant evaluation quantities without needing to
> recover the true (or normalized) reward or value functions.
>
> Finally, our results make explicit the structure of the softmax IRL solution
> set: Lemma 1 provides the Bellman equation satisfied by the behavior policy;
> Lemma 2 characterizes the full potential-based invariance; and Theorem 2 shows
> how imposing a normalization selects a unique representative. To our knowledge,
> this complete characterization of the solution set and the role of
> normalization has not appeared in prior IRL work. Deep PQR contains a lemma
> similar in spirit to our invariance result, but does not analyze the broader
> structure of the solution set or how different normalization choices correspond
> to different representatives.
>
>
> ---
>
> ## 2. Algorithmic differences
>
> We appreciate the reviewer noting the algorithmic similarity: both approaches
> estimate the behavior policy and then solve a linear fixed-point equation(s).
>
> Deep PQR recovers the reward by solving two coupled fixed-point equations: one for an
> anchored $Q$-function (under the anchor-action constraint) and a second equation
> that recovers the reward from this anchored $Q$-function. In contrast, our method
> requires solving only a **single** linear fixed-point equation, with both the reward
> and the associated soft Bellman solution obtained in closed form. This yields a
> substantially simpler algorithmic structure that is easier to analyze and introduces
> fewer sources of error.

---

> > ### Author Response · Authors · 2025-11-20
> >
> > ## 3. Our theoretical analysis substantially improves upon Deep PQR
> >
> > We appreciate the reviewer noting the algorithmic similarity: both approaches
> > estimate the behavior policy and then solve a linear fixed-point equation.
> > We will certainly appropriately cite and credit Deep PQR and draw the connections.
> > Our contributions **differ, however, in both assumptions and statistical guarantees**.
> >
> > **Known transition kernel.**
> > With a known transition kernel, we provide finite-sample error bounds that
> > characterize how error depends on the number of iterations, the discount factor,
> > and the generalization error of the estimated policy (Section 5.1). By contrast,
> > Theorem 1 of Deep PQR shows exact reward recovery only when *both* the behavior
> > policy *and* the transition kernel are known. It does not analyze
> > finite-iteration error or policy-estimation error and therefore does not yield
> > comparable finite-sample guarantees. Such results for known transition kernels
> > are also practically important—particularly in robotics and simulation-based
> > settings, where algorithms can be evaluated in a simulator or virtual
> > environment.
> >
> > **Unknown transition kernel.**
> > Our analysis improves on Deep PQR in several ways:
> >
> > - **Assumptions on policy estimation.**
> >   Deep PQR assumes a *sup-norm* error bound on the policy estimator (Assumption 2),
> >   which is often overly conservative and suffers severely under large action sets
> >   or is just generally unavailable in continuous
> >   state spaces. In contrast, we assume a high-probability *KL divergence*
> >   bound (Assumption 1), which is equivalent to excess log-likelihood and thus
> >   directly controlled by minimizing cross-entropy over a potentially nonparametric
> >   function class, opening the door to more practical algorithms.
> >   In particular, standard statistical learning theory directly
> >   yields these bounds. We currently give citations to such statistical learning results.
> >   In the revision we will add an appendix section reviewing these results more fully to show they
> >   directly give estimators, including neural net classifiers, that satisfy our assumptions.
> >
> > - **Model-agnostic generalization guarantees.**
> >   Deep PQR’s results are restricted to a very specific two-layer ReLU architecture and
> >   additional assumptions on width, initialization, and gradient-descent steps.
> >   In contrast, we derive generic guarantees for any given nonparametric function class,
> >   depending on a generic measure of function complexity.
> >   In particular, we give *PAC-style* regression bounds for each Bellman update
> >   (Lemma 6) using the *Rademacher complexity* of the regression class. This
> >   yields explicit \(L^2\) bounds for both the fixed point and the reward
> >   (Theorem 4; Lemma 5) that apply to *any* supervised learner and make error
> >   propagation across iterations transparent. We currently give citations on bounds
> >   on this complexity for a variety of function classes (including neural nets).
> >   In the revision we will add an appendix section reviewing implications for
> >   a variety of function classes by simply plugging in their complexity measure into
> >   our results.
> >
> > - **Correctly handling Bellman completeness.**
> >   Our analysis leverages a *Bellman completeness* condition to ensure
> >   realizability holds for each iteration of the regerssion step. Without
> >   this, one cannot ensure good estimation of the Bellman update iteration.
> >   Deep PQR appears to not have this assumption or something similar and,
> >   unless we misunderstood something, appears to incorrectly argue the
> >   convergence of their regression step. It is known that Bellman completeness
> >   is needed for FQI and we cited work showing exponential lower bounds
> >   if it is replaced by mere realizability of the true value function alone
> >
> > - **Correctly handling dependence across iterations.**
> >   Our analysis uses *sample splitting* across Bellman updates and properly
> >   conditions on the previous iterates in our iteration procedure
> >   in order to derive our high-probability guarantee,
> >   following standard analysis of fitted Q-iteration. Deep PQR
> >   updates all iterates on the same $n$ data points without
> >   accounting for the dependencies across iterations,
> >   making the validity of their proof unclear since
> >   it does not properly condition on estimated value functions
> >   from past iterations.
> >
> >
> > Overall, while the fixed-point idea also appears in Deep PQR, our work develops
> > a more general normalization framework, clarifies the partial-identification
> > structure relevant for value estimation, and provides a statistically grounded
> > finite-sample analysis with model-agnostic \(L^2\) guarantees. We will revise
> > the paper to appropriately cite Deep PQR and give it credit for the fixed-point
> > formulation, while articulating the differences above and our contributions, which
> > build upon Deep PQR.

---

### Author Response · Authors · 2025-11-20
**General Response**

We thank the reviewers for their constructive feedback. Two reviews highlighted
the importance of positioning our contributions relative to prior IRL work. We thank them for these pointers and will add a clear discussion of connections and
differences, which we overview below and give even more details on in our direct
responses to each reviewer.

**Relation to Deep PQR (Geng et al., 2020).**

*Overall:* Deep PQR also recognizes a fixed-point structure in MaxEnt IRL. Our
contribution is to make explicit the identification and partial identification structure and its implications for
policy evaluation, to provide a more general normalization framework, to simplify the fixed-point formulation to a
single linear equation, and to give a correct and model-agnostic analysis of finite-sample
guarantees on \(L^2\) error.
We will revise the paper to appropriately cite Deep PQR and give it credit for the fixed-point
formulation, while articulating the differences and our contributions, which
build upon Deep PQR.

Deep PQR focuses on recovering rewards under an anchor-action constraint via a
fixed-point update. Building on this insight, our paper develops several contributions that extend beyond what is available in Deep PQR or prior MaxEnt/DDC
IRL work:


- **Clarifying the identification structure.**
  We show how three ingredients fit together:
  (i) the fact that the log behavior policy satisfies the MaxEnt optimization
  problem,
  (ii) the potential-based invariances that cause non-identifiability, and
  (iii) how a chosen normalization selects a unique reward in the equivalence
  class.

  These ingredients appear implicitly across prior MaxEnt/DDC IRL work, but they
  have not been presented together in a unified way. Our contribution is to make
  their interaction explicit and show how they yield a simple linear fixed-point
  equation.



- **General normalization formulation.**
  We allow *any* linear normalization $ \mu r = 0 $, with action anchoring
  (fixing the reward of one action) as one example and room for
  domain-specific choices. For example, one might have prior knowledge that the
  reward is zero for a particular action (e.g., an outside option in
  econometric/dynamic discrete-choice models), or one may fix the expected
  reward under a given policy. This clarifies how normalization resolves reward
  non-identifiability and how different choices select different valid
  representatives within the equivalence class. It also suggests the possibility
  of future extensions that incorporate nonlinear or other domain-informed
  constraints.  In the revised version, we explain how our approach can be easily modified to
  impose a state-only baseline such as $ (\mu r)(s) = f(s) $ for a known
  function $ f $.



- **Partial identification and normalization-invariance of value evaluation.**
  We show that when the discount factor is known, policy value differences—and thus
  policy evaluation and comparison—are invariant to the choice of normalization. In
  these settings, recovering a normalized reward is unnecessary, as the log
  behavior policy alone identifies the relevant value functions. To our knowledge,
  this observation has not been stated explicitly in prior IRL work.

- **Algorithmic differences.**
  Deep PQR recovers the reward by solving two coupled fixed-point equations: one
  for an anchored $Q$-function (under the anchor-action constraint) and a second
  equation that recovers the reward from this anchored $Q$-function. In contrast,
  our method requires solving only a **single** linear fixed-point equation,
  providing the reward and associated solution to the soft Bellman equation in
  closed form. This yields a substantially simpler algorithmic structure that is
  easier to analyze and introduces fewer sources of error.

- **Correct and general finite-sample analysis.**
  Beyond providing a simpler algorithm, we provide finite-sample analysis that importantly differs on a few points

  (a) **Control \(L^2\)** Our bounds are on the \(L^2\) error, as is standard for function approximation.

  (b) **Model-agnostic** Our bounds hold for any classifier or regressor via dependence on their Rademacher
  complexity, a standard measure of capacity that is well understood for a wide range of classes.

  (c) **Separate classification and regression errors** Our bounds track how error depends on both policy-estimation accuracy and sample-based Bellman updates, as well as iteration error and approximation error.

  (d) **Accounts for dependence across iterations** Our high-probability analysis appropriately conditions on the dependence of the regression target in iteration $k$ on the model fit in iteration $k-1$.

  (e) **Accounts for realizability across iterations** We introduce a new adaptation of the standard Bellman
  completeness condition from offline RL to the task of solving IRL with operator iterations.


(continued in next comment)

---

> ### Author Response · Authors · 2025-11-20
>
> In contrast, Deep PQR provides finite-sample bounds only for its anchored reward
> estimator under a specific two-layer ReLU architecture, the anchor-action
> constraint, and unknown transitions. These guarantees rely on strong and nonstandard
> assumptions—including a sup-norm requirement on the policy estimator and
> conditions on network width, initialization, and gradient-descent steps.
>
> Additionally, the algorithm and analysis in Deep PQR appear to have a couple potential technical flaws.
> There appears to be no condition guaranteeing the realizability (potentially approximate)
> of their neural net model for each iteration. On the other hand we proposed a novel condition
> that adapts the classic Bellman completeness from offline RL, where it is known that Bellman completeness
> is needed for FQI and we cited work showing exponential lower bounds if it is replaced by mere realizability of the true value function alone.
> Additionally, Deep PQR updates all iterates on the same $n$ data points without
> accounting for the dependencies across iterations, making the validity of their proof unclear.
> We use sample splitting in order to be able to correctly condition on past iterated and obtain bounds,
> and we carefully balance the loss due to fewer iterations with the loss
> of splitting the data too finely.
>
> Moreover, Deep PQR does not provide finite-sample statistical guarantees for the
> known–transition-kernel setting, which is important in robotics and
> simulation-based applications. By contrast, our analysis yields
> model-agnostic \(L^2\) bounds for a single fixed-point update, holds for any
> supervised learner via Rademacher-complexity arguments, and applies in both
> known and unknown transition-kernel regimes.
>
>
>
> **Relation to Fu et al. (2018)**
>
> Theorem~1 of Cao et al.\ (2021) shows that many rewards can rationalize a given
> policy under an entropy-regularized MDP, characterizing an equivalence class of
> solutions. Their focus is on partial identifiability, rather than on how a
> normalization constraint (e.g., \( \mu r = 0 \)) can be used to select a unique
> representative or on developing an associated estimation procedure. We will make
> sure to cite this work and discuss it.
>
> **Relation to Cao et al. (2021).**
>
> AIRL (Fu et al., 2018) also notes that the log behavior policy satisfies the
> MaxEnt optimality relation, although this observation does not play a direct
> role in their theory or algorithm. Instead, they address non-identifiability by
> restricting to state-only rewards and by developing an adversarial min–max
> training loop. In contrast, we consider general state–action rewards, impose an
> explicit normalization to select a unique representative, and show that this
> structure yields a simple linear fixed-point equation and a straightforward
> classification–regression algorithm. We already cite AIRL, and we will expand
> our discussion thereof with these additional points.
>
> **Relation to Zeng et al. (2023).**
>
> Zeng et al. (2023) learns a **generative world model**—e.g.,
> a diffusion-based model of the transition dynamics—and then
> optimize a maximum-likelihood or adversarial objective over trajectories. This
> requires training and differentiating through a high-capacity generative model
> and solving a substantially more complex optimization problem. Our approach avoids
> generative modeling entirely. By reducing IRL to
> **classification plus regression** (of just conditional means, no distributions need be
> learned), we obtain a closed-form fixed-point
> structure and transparent statistical guarantees. We will incorporate this
> comparison into the related-work section.

---

### Author Response · Authors · 2025-12-01
**Summary for the AC**

Summary for the AC

Our paper revisits MaxEnt / softmax IRL and dynamic discrete choice (DDC) and shows that the population maximum-likelihood solution is characterized by a single linear fixed-point equation involving the behavior policy, develops a simple and modular “classify–then–regress” procedure, and provides strong model-agnostic finite-sample guarantees. In the original pre-discussion reviews, two reviewers were marginally above acceptance (6, 6), one was marginally below but open to acceptance (4), and one was negative (2). The positive and borderline reviews consistently praise the simplicity of the reduction, the clean connection to DDC, and the thorough analysis; remaining concerns involve normalization, positioning relative to recent IRL/identifiability work, and the scope of experiments. Below we summarize these issues and how we addressed them, emphasizing the additional clarity and contributions that emerged during the discussion.

1. Normalization, identifiability, and meaning of the recovered reward

**Concern.** Several reviewers worried that our normalization condition $\mu r = 0$ looked like a strong assumption on the true expert reward, or like selecting an arbitrary reward in the equivalence class. One reviewer (HuNX, score 2) focused heavily on whether our particular normalized reward has a special status.

**Our response and contribution.**
We clarified that we do not assume the expert’s true reward satisfies $\mu r = 0$. As in the identifiability literature, the reward in entropy-regularized IRL / DDC is determined only up to potential-based shaping. Our normalization is explicitly framed as an analysis device for selecting one representative from this equivalence class; any such representative rationalizes the same behavior and yields the same policy value differences.

We showed that our framework is more general than anchor-action: anchor actions and uniform “sum-to-zero’’ constraints arise as special cases of our linear normalization.

We emphasized a concrete structural result: policy value differences and policy comparison are invariant to the normalization when $\gamma$ is known. Thus, for many IRL goals (policy evaluation, comparing policies), the log behavior policy already suffices; the normalization affects only which structural reward is reported, not which policies are preferred.

We explained how domain knowledge enters through $\mu$: in DDC, a zero-utility outside option; in robotics, a no-op or neutral baseline action. Our framework cleanly encodes such choices and thereby recovers the domain-meaningful reward within the equivalence class.

After rebuttal, the positive reviewers did not contest our clarifications and kept their positive scores; they accept that a normalization is unavoidable and that our framework subsumes standard choices. The negative reviewer continues to question whether our normalization is uniquely compelling without more real-world illustrations, but acknowledges that some criterion is necessary. Indeed, a normalization is required, and ours is no less principled than existing ones.

---

---

> ### Author Response · Authors · 2025-12-01
>
> 2. Relation to prior work and novelty (Deep PQR, AIRL, Cao/Skalse, offline IRL)
>
> **Concern.** Reviewers asked how our structural results differ from Deep PQR and from identifiability work such as AIRL, Cao et al., and Skalse et al., and noted that some facts (e.g., log-policy as a solution) are known.
>
> **Our response and contribution.**
> We explicitly credit prior work: Lemma 1 (log policy as a trivial solution) is not claimed as a new theorem; we now present it as a simple, known identity included to make the argument self-contained.
>
> **Compared to Deep PQR, we stressed:**
>
> - We strictly generalize their anchor-action setting via our normalization framework, characterize the partial identification set, and show that policy comparison is identified without normalization.
> - Algorithmically, we solve one linear fixed point rather than two coupled ones, yielding a simpler iterated regression algorithm.
> - Our finite-sample analysis is stronger:
>   - It is model-agnostic, accepting any function class for value functions and expressing bounds in standard complexity measures (rather than a very specific neural architecture in Deep PQR).
>   - It flexibly handles behavior policy estimation because it depends on excess log-likelihood, obtainable from classic M-estimation results (rather than needing unrealistic $L^\infty$ guarantees).
>   - It is stated in $L^2$ error on value functions, consistent with learning-theory results and suitable for downstream tasks (rather than weaker $L^1$ guarantees).
>   - It correctly handles statistical dependence across iterations using sample splitting and choosing the number of iterates to balance sample sizes and fixed-point-approximation error (rather than omitting conditioning on past iterates).
>   - It correctly enforces realizability at each iterate by adapting Bellman completeness to our IRL problem (rather than omitting an explicit condition).
>
> **Compared to AIRL and Cao/Skalse:**
>
> - AIRL uses a related identity inside a min–max loop and resolves non-identifiability via state-only rewards; we allow general state–action rewards, apply explicit normalization, and obtain a closed-form fixed point plus a simple classification–regression algorithm.
> - Cao, Skalse, and related works characterize which rewards belong to the equivalence class; we build on that viewpoint to select and estimate a unique normalized element with finite-sample guarantees.
>
> We also added a comparison to offline IRL with generative models (e.g., Zeng et al.), noting that our method avoids training generative world models and reduces IRL to regression plus classification with a transparent error decomposition.
>
> After our responses, the technically positive reviewers did not maintain concerns about novelty; they continue to view the work as offering an interesting structural perspective and a clean, analyzable algorithm.
>
> ---
>
> 3. Theoretical assumptions and connection between deterministic and sample-based analysis
>
> **Concern.** Reviewers asked how our deterministic fixed-point analysis relates to the sample-based algorithm, and how realistic Assumptions 1 (KL control) and 2 (Bellman completeness) are.
>
> **Our response and contribution.**
> We clarified that Lemmas 3–4 and Theorem 3 are population statements for the operator $T_u$. Section 5 then shows how a fitted-iteration scheme with sample splitting tracks this contraction, in line with modern fitted Q-iteration analysis.
>
> **Assumption 1** is a standard PAC guarantee for probabilistic classification: excess cross-entropy / KL is controlled by learning-theory tools for many function classes, including neural nets; these guarantees follow from classic M-estimation results (e.g., van de Geer). This KL error appears additively in reward and value errors, scaled by $(1-\gamma)^{-1}$.
>
> **Assumption 2** (Bellman completeness) ensures realizability of each regression iterate. It is analytic and does not change the algorithm. Some realizability is necessary. It can be relaxed to approximate completeness with the approximation error included in the final bounds. Recent work in offline RL shows such an assumption is needed to estimate value functions without exponential dependence on horizon.
>
> Reviewer HUTV wrote, after reading our rebuttal, that “the analysis section seems fine” and maintained their positive score; remaining reservations concern scope rather than correctness.

---

> ### Author Response · Authors · 2025-12-01
>
> ---
>
> 4. Experiments and scalability
> **Concern.** Experiments are on gridworlds only; reviewers asked whether the method scales and how it compares to GAIL/AIRL/IQ-Learn on standard continuous benchmarks.
> **Our response and contribution.**
> We clarified that our focus is analytical, and the experiments are deliberately proof-of-concept—designed to validate the reduction and to illustrate that, even with simple neural approximators, our method matches or outperforms MaxEnt IRL, especially in the misspecified setting (where we significantly improve both reward recovery and policy matching).
>
> We emphasized that scalability is inherited from standard supervised-learning components: step 1 is any probabilistic classifier; step 2 is a fitted value-iteration loop via regression. Our finite-sample guarantees are stated in terms of general complexity measures, making the method directly compatible with modern neural architectures used in large-scale RL.
>
> We explained that our Bellman updates are on-policy with respect to the expert data and evaluated in an \(L^2(\lambda)\) norm tied to the behavior stationary distribution, which avoids the extrapolation pathologies common in offline RL with OOD queries.
>
> The positive reviewers accepted this positioning (theoretical focus + proof-of-concept experiments) and kept their scores, while noting that broader empirical studies would be a natural next step.
>
> ---
>
>
> 5. Post-rebuttal status and recommendation
> Three reviewers (original scores 6, 6, 4) now view the technical content as sound; two explicitly praise the clarity and “nice and sweet” analysis.
>
> The remaining negative reviewer’s objections concern how compelling our normalization is without additional real-world experiments, rather than correctness of the structural results or guarantees. However, as the other reviewers recognize, a normalization is necessary to select a representative from the observational equivalence class induced by state-based reward shaping, and ours provides a general statement that includes standard normalizations used in prior work.

---

### Meta-Review · Area_Chair_7G3P · 2026-01-04

**Summary:**

Across reviews, there was broad agreement that the paper is technically solid and presents a clean and elegant reformulation of MaxEnt IRL as a classification-plus-regression problem. Several reviewers appreciated the simplicity, the connection to dynamic discrete choice, and the careful finite-sample analysis.

However, the key concern that ultimately drives my decision is conceptual rather than technical: whether the specific normalized reward recovered by the method is meaningful or useful beyond enabling the fixed-point formulation. One reviewer in particular consistently argued that the normalization appears arbitrary without clear domain grounding or convincing empirical evidence, and that the paper does not demonstrate why this reward should be preferred over other equivalent representatives. In addition, multiple reviewers noted that the experimental evaluation is limited to small gridworlds and does not convincingly establish practical relevance or scalability, especially given the strong dependence on accurate behavior-policy estimation.

**Reviewer Concerns:**

The rebuttal did a good job clearing up most of the technical questions and misunderstandings. It made it clear that the normalization is a choice rather than an assumption about the expert’s true reward, clarified how the method relates to Deep PQR, AIRL, and identifiability work, and explained the link between the population fixed-point analysis and the sample-based algorithm, as well as the role of the assumptions in the theory. What remains unresolved is more conceptual than technical: at least one reviewer is still not convinced that the particular normalization leads to a reward with meaningful interpretation beyond enabling the fixed-point formulation, especially without stronger empirical evidence. The limited gridworld experiments and the reliance on accurate behavior-policy estimation also continue to raise doubts about practical relevance and scalability.

**Reviewer Scores:**

If reviewers had been able to update their scores, I expect the two reviewers who were already positive (both at 6) would have stayed at roughly the same level, but with higher confidence after the clarifications. The reviewer who was initially borderline (4) would likely have moved up modestly, as most of their technical and novelty concerns were addressed. In contrast, the strongly negative reviewer (2) would likely not have changed their score, since their core objection about the arbitrariness and practical meaning of the normalization was not resolved to their satisfaction.

---

### Decision · Program_Chairs · 2026-01-26

Reject